# Snow depth on Arctic sea ice from historical in situ data

Elena V. Shalina[1,2], Stein Sandven[3]

[1]Nansen International Environmental and Remote Sensing Centre, St.Petersburg, 199034, Russia
[2]St. Petersburg State University. Institute of Earth Sciences, St.Petersburg, 199034, Russia
[3]Nansen Environmental and Remote Sensing Center, 5006 Bergen, Norway

*Correspondence to*: Elena Shalina (elena.shalina@niersc.spb.ru)

**Abstract.** The snow data from the Soviet airborne expeditions Sever in the Arctic collected over several decades in March, April and May have been analyzed in this study. The Sever data included more measurements and covered a much wider area, particularly in the Eurasian marginal seas (Kara Sea, Laptev Sea, East-Siberian Sea and Chukchi Sea), compared to the
Soviet North Pole drifting stations. The latter collected data mainly in the central part of the Arctic Basin. The following snow parameters have been analyzed: average snow depth on the level ice (undisturbed snow) height and area of sastrugi, depth of snow dunes attached to ice ridges and depth of snow on hummocks. In the 1970s-80s in the central Arctic the average depth of undisturbed snow was 21.2 cm, the depth of sastrugi (that occupied about 30% of the ice surface) was 36.2 cm and the average depth of snow near hummocks and ridges was about 65 cm. For the marginal seas the average depth of
undisturbed snow on the level ice varied from 9.8 cm in the Laptev Sea to 15.3 cm in the East Siberian Sea, which had a larger fraction of multiyear ice. In the marginal seas the spatial variability of snow depth was characterized by standard deviation varying between 66 and 100%. The average height of sastrugi varied from 23 cm to about 32 cm with standard deviation between 50 and 56%. The average area covered by sastrugi in the marginal seas was estimated to be 36.5% of the total ice area where sastrugi were observed. The main result of the study is a new snow depth climatology for the late winter
using data from both the Sever expeditions and the North Pole drifting stations. The snow load on the ice observed by Sever expeditions has been described as a combination of the depth of undisturbed snow on the level ice and snow depth of sastrugi weighted in proportion to the sastrugi area. The height of snow accumulated near the ice ridges was not included in the calculations because there are no estimates of the area covered by those features from the Sever expeditions. The effect of not including that data can lead to some underestimation of the average snow depth. The new climatology refines the
description of snow depth in the central Arctic compared to the results by Warren et al. (1999) and provides additional detailed data in the marginal seas. The snow depth climatology is based on 94 % Sever data and 6 % North Pole data. The new climatology shows lower snow depth in the central Arctic comparing to Warren climatology and more detailed data in the Eurasian seas.

## 1 Introduction

Most of the Arctic sea ice is covered with the snow year round except in the melt season when meltponds are present. Snow cover plays an important role in the thermodynamic processes of sea ice. In winter the snow ensures high sea ice surface albedo of about 0.85 (Perovich and Polashenski, 2012) associated with low energy absorption. On the other hand, snow insulates the sea ice from the influence of cold air and reduces the rate of ice growth. After snow begins to melt in summer producing melt ponds, reduction of the surface albedo and higher energy absorption results in a more rapid ice melt. Thus data on snow depth and surface albedo is important for quantification of the thermodynamical processes. Besides, information on snow depth is also very important for ice thickness retrieval from satellite altimeter measurements of sea ice freeboard and their conversion to thickness using hydrostatic equation (Laxon et al., 2013, Kwok et al., 2017, Kern et al., 2015).

Changes of the arctic ice cover over the last decades are well documented in data from different satellite, aircraft, submarine, buoy and in situ measurements. There is clear evidence of a decline in sea ice area and thickness (Kwok and Rothrock, 2009; Comiso and Nishio, 2008; Wadhams, 2012; Stroeve et al., 2012; Lindsay and Schweiger, 2015). This implies that the Arctic sea ice changes from predominantly multiyear ice to increased fraction of seasonal ice (Maslanik et al., 2007; Tschudi et al., 2010, Maslanik et al., 2011; Comiso, 2012; Tschudi et al., 2016). As a result the whole ice pack becomes more vulnerable to strong atmospheric impacts (Parkinson and Comiso, 2013). Sea ice retreat leads to larger areas of open ocean, which absorbs more solar energy and consequently enhances the warming of the upper layer of the Arctic Ocean. This warming also contributes to melting of the sea ice underside (Perovich et al., 2007). Reduction of the sea ice cover also amplifies warming of the atmospheric boundary layer in the high latitudes (Screen and Simmonds, 2010). This process may accelerate the sea ice decline and diminish the proportion of precipitation in form of snow (Screen and Simmonds, 2012).

In situ observations of the snow cover of the Arctic sea ice are presently very scarce, especially year-round measurements which are needed to document the seasonal variability of the snow cover. The most extensive data set in the past was collected during the Soviet North Pole (NP) drifting stations in 1937 and 1954-1991. Data from these expeditions have been used to establish the Warren snow climatology data set (Warren et al., 1999), hereafter denoted W99, providing distribution of the snow depth and density for each month of the year. Valuable data on snow properties has also been collected from other expeditions, buoy measurements, ice camps and validation experiments in specific areas of the Arctic. In situ snow depth measurements has been carried out in spring time in the Beaufort Sea, Elson Lagoon and Chukchi Sea (Sturm et al., 2002, Sturm et al., 2006, Markus et al., 2006, Newman et al., 2012, Nghiem et al, 2013, Webster et al, 2014), in the Canadian Arctic between the islands (Iacozza and Barber, 1999, King et al., 2015) and near the coast of northern Greenland (Farrel et al., 2012, King et al., 2015). With the aim to describe spatial distribution of snow on sea ice as a function of ice type, field experiments SIMMS'95 and C-ICE'96 in the Canadian Arctic have been conducted (Iacozza and Barber, 1999). Observations were made at the sites covered with only one sea ice type and snow topography on a given type of ice was described using variogram modelling. In 1997-98, extensive snow depth measurements have been made during SHEBA

(Sturm et al., 2002). One of the objectives was to record temporal evolution of snow depth over the year, to evaluate its spatial variability (as broad as conditions of the experiment allowed), and to estimate the freshwater amount contained in the snow cover. Changes of snow depth connected with ice type and the level of deformation were also studied under SHEBA. Another expedition, the AMSR-Ice03 validation campaign carried out in March 2003 offshore of Barrow collected snow

data for comparison with satellite products and also for analysis of snow depth on the sea ice of different age and different roughness (Sturm et al., 2006, Markus et al, 2006). During the Norwegian Young Sea ICE (N-ICE2015) campaign, snow depth was measured on the sea ice north of Svalbard (Gallet et al., 2017, Merkouriadi et al., 2017). In recent years a series of IceBridge validation/calibration campaigns have been conducted including in situ snow measurements on several ice types: undeformed level first-year (FY) ice, multiyear (MY) ice, and heavily deformed pressure ridges. Results have been

published from studies near Greenland in April 2009 where measurements collected at the Danish GreenArc sea ice camp (Farrell et al., 2012), from surveys in the Beaufort Sea in March 2011 (Gardner et al., 2012; Newman et al., 2014), from measurements taken in March 2012 during BROMEX field campaign (Webster et al., 2014) and in March-April 2014 in the Canadian Arctic Archipelago waters close to Eureka and in the Lincoln Sea near the northern coast of Greenland (King et al., 2015). Assessment of five snow depth retrieval algorithms that differently process IceBridge snow radar data has been made

through comparison with field measurements from two ground-based campaigns, 2012 BROMEX near Barrow, Alaska, and 2014 Eureka, near the research base with the same name in Canada (Kwok et al., 2017).

The W99 snow climatology provides monthly averaged gridded snow depth maps for the whole Arctic, representing the means over the whole period of the North Pole (NP) drifting station observations. However, the averages are based on measurements from usually not more than two stations established in the Arctic in any given year. Due to the high spatial

and temporal variability of the snow depth it is difficult to adequately estimate the uncertainty of the mean values of the W99 climatology. Another serious limitation of W99 climatology is the fact that the NP drifting stations were established on the MY ice, which means that climatology does not include snow on FY ice and is therefore heavily biased towards MY ice.

In this paper, we analyze snow data from Soviet airborne expeditions Sever that was collected through 28 years in the middle of the NP time period and cover much wider area than NP stations. The Sever expedition landings were made not only on the

MY ice in the central Arctic but also on the FY ice in the Eurasian seas, especially on the Siberian shelf where practically no NP data was collected. The main goal of this study is to produce improved data set of snow depth for the whole Arctic for the late winter season (March-April-May) by combining Sever and NP data sets. Both data sets were collected mainly in the 1960s, 70s and 80s. The combined snow distribution data set and obtained dependencies will be useful for validating sea ice and climate models and also as input into the algorithms retrieving ice thickness from satellite altimeter data. The data set

will be important for comparison with snow observations in the present sea ice conditions.

The snow depth varies significantly within the Arctic region. It varies in time (both seasonally and interannualy) and space (over large distances and locally within a single ice floe). Ice age and intensity of precipitation are the main factors to determine the snow depth. The older the ice is the more snow can accumulate depending on the amount of precipitation. Furthermore wind and ice roughness play a role in determining the distribution of the snow depth. Fresh snow on an ice floe

can easily be blown away from the smooth ice into the rough ice areas. On the rough ice the blowing snow is trapped and consequently snow depth is larger in the areas of ice ridges and hummocks in comparison with the smooth ice area. Another manifestation of snow depth deviation caused by wind is sastrugi, irregular ridges and grooves of snow formed on the ice surface. When the snow is light, snow dunes are easily moved by the wind, however they consolidate by the end of winter due to compression and crystallization. All mentioned aspects of snow depth variations in the end of winter season are described in this paper basing on Sever expeditions measurements.

This paper is organized as follows: in Sect. 2 we describe the data collected during Sever expeditions and compare it briefly with NP data. In Sect. 3 we describe methods used for data processing. Description of the depth of snow cover atop Arctic sea ice of different roughness is given in Sect. 4. That section also contains separate description of snow on fast ice, results on combination of Sever and NP data and analysis of contemporary buoy snow depth measurements. Discussion of the new results presented in the paper is provided in Sect. 5.

## 2  Data Description

The airborne Sever expeditions took place in 1937, 1941, 1948-1952, and 1954-1993. The first expedition was organized to support the Soviet drifting station North Pole-1 (NP-1). The personnel, goods and equipment were transported by the Sever expedition airplanes to the ice floe at  89° 25′ N, 78° 40′ W where NP-1 was deployed. The NP-1 expedition collected different oceanographic, meteorological and gravimetric measurements during its 9 month successful operation. The valuable experience of landing an aircraft on sea ice laid the foundation for further Arctic airborne expeditions. It was decided that oceanographic, meteorological, snow and ice measurements in the Arctic should be done by ice researchers throughout a series of short landings of specially equipped airplanes. The advantage of such observations was that a wide area of the Arctic could be covered by landings and the sites to visit could be chosen in accordance with the goal of the study.

Most of the landings took place from mid-March to early May, when there was enough daylight to operate and before melting started and aircraft could not land on the ice. In some years landings occurred also before March and after May (Table 1). In contrast to the NP data, which covers only MY ice, the Sever data were collected on both MY and FY ice, as long as the ice could provide a runway for the aircraft.

**Table 1**. The number of landings of Sever expeditions by decade and month.

|  | January | February | March | April | May | June | July |
|---|---|---|---|---|---|---|---|
| **1930s** |  |  |  | 2 | 5 |  |  |
| **1940s** |  |  |  | 21 | 24 |  |  |
| **1950s** |  | 2 | 21 | 286 | 147 | 4 | 1 |
| **1960s** | 3 | 13 | 107 | 282 | 290 | 46 |  |

| | | | | |
|---|---|---|---|---|
| **1970s** | | 438 | 679 | 148 |
| **1980s** | 3 | 380 | 526 | 339 |

The Sever data used in this study were obtained from the US National Snow and Ice Data Center (NSIDC) (http://nsidc.org/data/g02140). The dataset contains sea ice and snow measurements of 23 parameters, in particular, including ice thickness and snow depth and density on the runway and surrounding area, as well as dimensions and snow coverage of ridges, hummocks, and sastrugi. Not all parameters were measured at every landing. Only ice thickness measurements were conducted over the whole period of Sever expeditions. Observations of snow started in 1959 and were conducted up to 1988 with small time gaps. The present study is concentrated on analysis of data from March, April and May (the MAM months further in the text) when most of the data was collected.

The monthly mean positions of the NP drifting stations in 1954-1991 for the MAM months and landing sites of Sever expeditions where snow measurements were conducted in 1959 -1986 in the same months are shown in Fig. 1. The NP data covers mainly the central part of the Arctic while the Sever data covers much larger areas with most of the data collected along the Siberian shelf seas between Novaya Zemlya and the Bering Strait. The selection of sites for the NP station deployment depended on requirements for sufficient ice thickness, floe size, possibilities for cargo aircraft landings and other factors.

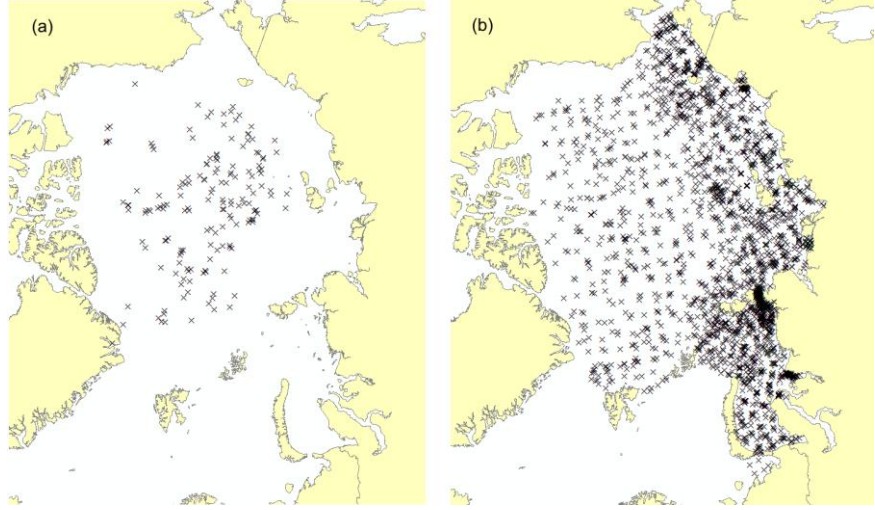

**Figure 1**. Monthly mean positions of NP drifting stations in 1954-1991 (a) and Sever expedition landings where snow measurements were conducted in 1959-1986 (b). Only observations in the MAM months are shown for both sets of measurements.

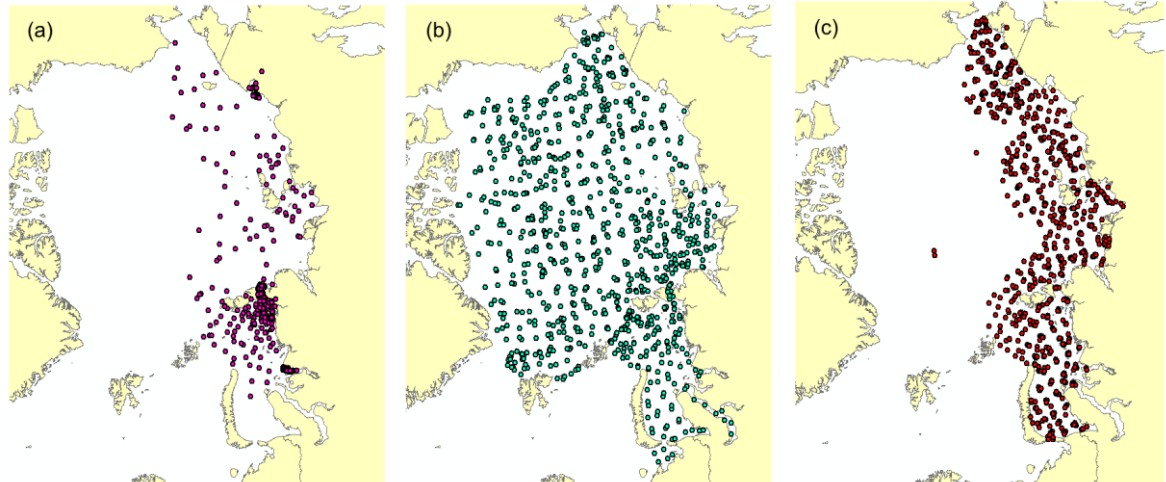

**Figure 2**. Sever expedition landings where snow measurements were conducted in the MAM months: 60s (a), 70s (b) and 80s (c).

The geographical distribution of the Sever landings for each of the three decades when most of the snow data was collected

is shown in Fig. 2. In the 60s (including 1959) the landings were focused in the Russian part of the Arctic Ocean with most frequent observations in the northern Kara Sea where 69 % of all landings took place. In the 70s the landings were distributed over most of the Arctic Ocean, while in the 80s observations were concentrated in the Siberian shelf seas, supporting the Northern Sea Route.

The Sever and NP expeditions had different data collection strategy not only regarding spatial and temporal sampling, but

also in the way snow observations were collected.

At the NP stations three types of snow measurements have been provided: snow line measurements, snow gauge precipitation measurements and snow stake measurements at the meteorological station site (Radionov et al. 1997; Colony et al, 1998). Snow line measurements were repeated once per month and sometimes once per 10-day period along the same lines of 500 m or 1 km length. The distance between each measurement (along the line) was 10 meters. The snow line was

selected on a flat ice surface with no human objects or ice hummocks that could influence the snow depth (Colony et al, 1998). The snow line surveys were carried out and documented if the average snow depth along the line was at least 5 cm. Snow cover and the ice under it changed during the station lifetime and line measurements allowed "to obtain a representative distribution of snow depths, passing through sastrugi, snow dunes, and pressure ridges as well as level snow" (Warren et al., 1999). The advantage of the NP line measurements is that natural local variability on the same ice is captured

as well as the time evolution over the year. Other measurements of snow depth were carried out daily using three permanent snow stakes installed at the meteorological station site. These sites were generally located close to the station camp, implying that observations could be influenced by camp structures. Consequently, the measurements of snow cover depth from snow stakes in many cases did not agree with the line measurements (Radionov et al., 1997). Also snow gauge measurements did

not always agree with the line measurements. W99 used mainly snow line measurements to produce their snow climatology data set considering those measurements most reliable.

The benefit of the Sever expeditions was that data were collected over much larger geographical area compared to NP data. During Sever landings various snow depth measurements were carried out in different locations on and around the runway. The measured parameters included runway snow depth, snow depth on the prevailing ice in the landing area, snow depth at mid-length of snow dunes extending out from ice ridges, depth of snow on hummocks, on both windward and lee sides, and height of sastrugi on the dominant type of ice in the landing area. Representative areas for measuring snow parameters were chosen from the air before landing, including estimation of sastrugi areas. The runway was chosen on flat ice that was most probably first year ice, but could also be multiyear ice. Meanwhile, the ice conditions around landing track were usually different from that on the runway: the difference between the ice thickness of the runway and of the area where other measurements were conducted was in some cases about 300 cm. In the description of data the ice in the area around runway is called "prevailing ice of the landing area". Later in the paper we use definition "undeformed ice" as a substitute for "prevailing ice of landing area", since snow cover associated with ice features caused by ice deformation like ice ridges and hummocks is described separately. After landing, snow depth was measured at 10-20 random points on prevailing ice of the landing area and on ice surface with distinctive features. For snow depth of more than 10 cm, at least 10 measurements were conducted over the entire ice floe, as well as on adjacent floes. How many measurements were made in case when snow depth was lower than 10 cm is not indicated in the description of data. The depth of snow dunes stretching from ice ridges and depth of snow on hummocks were measured using the following steps. The snow depth on 2 or 3 snow-covered hummocks was measured on both windward and lee sides at 10-20 points. The depth of snow dunes stretching from ice ridges were measured at 3-5 sites at their mid-length. The height of sastrugi on the undeformed ice was measured at several points. Note that all types of snow dunes formed on a flat ice surface by wind were referred to as sastrugi in Sever expeditions' data set. The averaged measurements of the mentioned parameters were reported in the documents from each expedition.

In addition to the description of Sever data measurements, it is important to mention that in all cases observations were conducted by highly experienced personnel who selected typical objects for measurements in order to produce a representative picture of the landing area. Estimating value of Sever observations, we can mention that data on distribution of sastrugi area over the Arctic has not been found in other studies. Data on sastrugi height and on snow attached to hummocks and ridges are very few.

Some statistics about the snow measurements of the Sever expeditions are shown in Figure 3. From 1959 to 1989 a total of 3234 landings were conducted, of which 2331 landings provided snow depth measurements. The number of landings increased significantly during the 1970s, associated with expanding the surveys to cover the whole Arctic Ocean (Fig. 2). Towards the end of the 1980s the Sever programme came to an end, with the last expedition in 1989. Altogether, most intense and broad snow measurements have been carried out in the period 1977-1986. In the paper we analyze all parameters shown in Fig. 3 except for one – area of hummocks that was measured only 555 times.

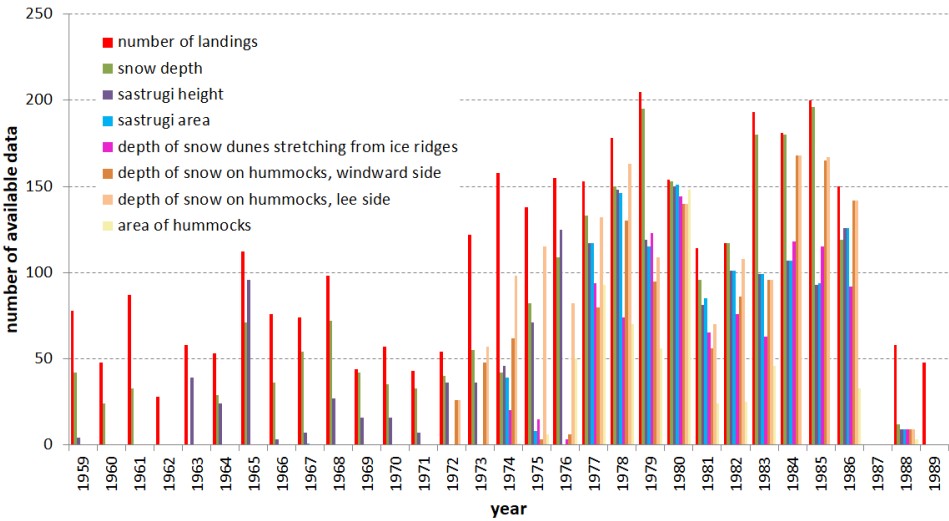

**Figure 3**. The number of landings and the number of different snow observations per year carried out in the MAM months.

Data on the depth of undisturbed snow measured on the prevailing type of ice in the landing area from all Sever expeditions landings over the period from 1959 to 1988 is presented in Fig. 4. Since a substantial portion of measurements was conducted on the FY ice the snow depth is below 20 cm for 79 % of all samples.

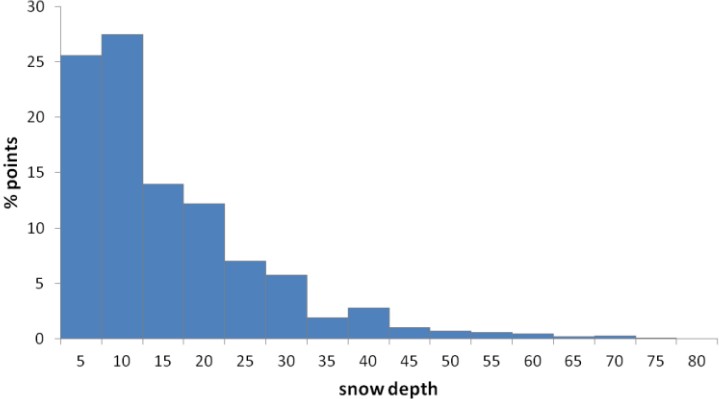

**Figure 4**. Observations of snow depth on prevailing landing area ice from Sever expeditions made in the MAM months, 1959-1988. Observations are grouped in 5 cm intervals.

## 3 Methods

The Sever data contains the following snow parameters: average snow depth of the undisturbed landing site, height and area of sastrugi (on the dominant type of ice in the landing sites), depth of snow dunes attached to ice ridges (at mid-distance between ridge and undisturbed snow area), and depth of snow on hummocks (on windward and lee sides). These snow

parameters are interconnected, and a statistical description of each of them can be useful in order to characterize snow cover on sea ice. The analysis also included preparation of a data set that can be compared and integrated with the NP data and thereby improve W99 the snow cover climatology.

The spatial variability of snow parameters in the MAM months is one of the important characteristics that can be elucidated using the Sever data. Snow depth variations on local scale are caused by wind forcing, resulting in sastrugi formation, increased snow depth near ice ridges and hummocks and reduced snow depth on level ice. On larger scale snow depth variations are synoptic in origin (Sturm et al., 2006) or caused by different age of ice where snow cover has been built up. Local and large scale variations are further discussed in Sect. 4.

The large scale spatial variability of snow depth is quantified by averaging observations from the landing sites in 100 by 100 km grid cells, as shown in Fig 5. To produce regular grids we averaged all points within the 3x3 cell neighborhood around every grid cell. The number of measurement sites used in the gridding operation varied from 1 to 23 in the Central Arctic and from 25 to 76 in the Siberian marginal seas (Kara Sea, Laptev Sea, East Siberian Sea and Chukchi Sea). The area with the highest density of measurements (around 160) lies in the north-eastern part of the Kara Sea and in the Vilkitsky Strait, which was the priority area for the Sever expeditions in the 1960s (Fig. 2a and 5b).

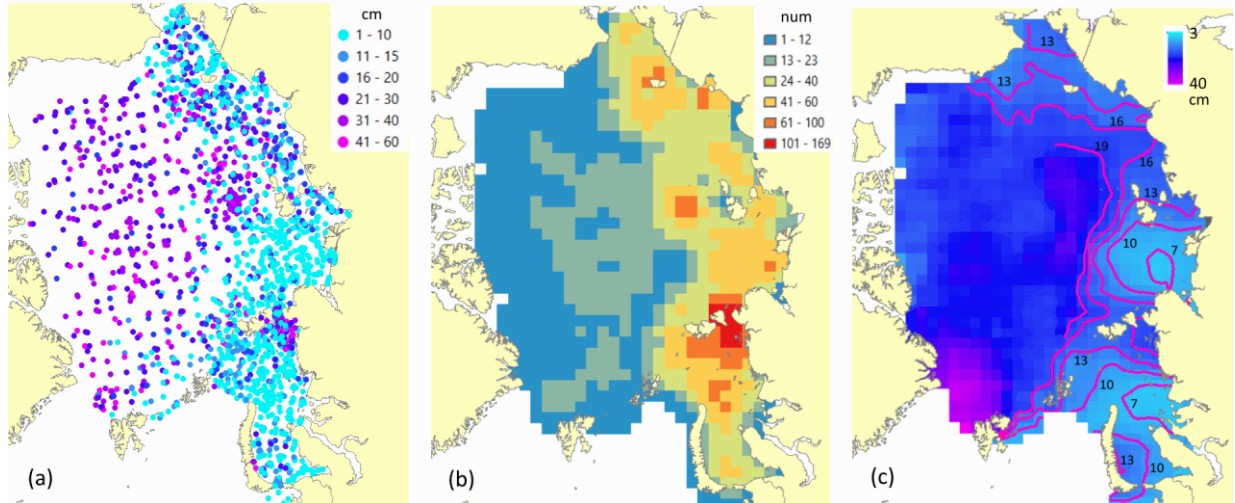

**Figure 5**. Spatial distribution of measured snow depth on the level ice around Sever landing sites, MAM months, 1959-1988 (a). Number of point measurements used in gridding, grid cell is 100x100 km (b). Contour lines of the snow depth in the marginal seas produced using observations shown on the left panel (c). Snow depth is in cm.

The Siberian marginal seas are poorly represented in the W99 climatology but quite well represented in the Sever data. Hence, the Sever data can provide a significant contribution to improved snow climatology in these areas as well as in the central part of the Arctic Ocean. The depth of undisturbed snow is typically less than 10 cm in the Kara and Laptev seas and less than 20 cm in other areas (Fig.5c). With included sastrugi, snow dunes and snow on hummocks, the average snow depth estimates become larger, as discussed in Sect. 4 and 5.

In the analysis it was useful to discriminate observations from FY and MY ice because MY ice as a platform to accumulate snow is different from FY ice. Firstly, MY ice exists from the very beginning of the fall and thus is able to catch the earliest falling snow. Secondly, the topography of MY ice is different from that of FY ice, being more irregular, that influences redistribution of snow. Furthermore, there are melt ponds on the MY ice in summer that become local depressions during

winter, accumulating snow and forming deeper than average snow patterns (Perovich et al., 2003). We separate observations from FY and MY ice by using an ice thickness threshold, since ice thickness was measured at every landing. The threshold was chosen to 2.0 m, implying that ice thinner than 2 m is defined as FY ice and ice thicker than 2 m is defined as MY ice (WMO Sea-Ice Nomenclature). In the decades of the Sever expeditions, MY ice dominated in the central Arctic, the fraction of MY ice being close to 100% in 60s-80s. Using the 2 m threshold, it was found that 78% of Sever observations from the

central Arctic were conducted on the MY ice and 22% on the FY ice. The sampling made by the Sever expeditions was probably biased towards level ice. In order to derive maximally representative data set for the central Arctic, we decided to use only MY-based measurements for that area. Supposing that NP and Sever observations complement each other we merged Sever snow depth observations from MY ice with NP data to produce snow depth climatology for the central Arctic. By using the same ice thickness threshold in the marginal seas, the fraction of data coming from MY ice was 9% in the Kara

Sea, 11% in the Laptev Sea, 34% in the East-Siberian Sea and 23% in the Chukchi Sea. These fractions of MY ice seem reliable, thus all snow observations in the marginal seas (both FY- and MY-based) were used in the subsequent analysis.

Snow data collected on the FY and MY ice were analyzed to find an empirical relation between snow depth and sea ice thickness. The relation is estimated from a regression analysis and is presented in Sect. 4. Such relation can be expected because both ice thickness and snow depth grow through the freezing season from September to May, although the relation

is not straight forward as snow cover acts as an insulator reducing the freezing rate of the ice.

In processing of the sastrugi data, average height and areas covered by sastrugi were estimated. Furthermore, attempts were made to find relation between sastrugi height and snow depth in the surrounding areas. Spatial changes of sastrugi height were identified through estimating averages for different Arctic regions. In addition to height, the spatial variability of the area of sastrugi was analyzed because it contributes to the estimation of the average depth of snow cover. Snow depth

associated with ridges and hummocks was also estimated; but the effect of this part of the snow cover on the averaged snow depth could not be evaluated because the areas covered by these features were not sufficiently observed.

The fraction of snow data collected on fast ice could be extracted using sea ice climatology data GO2172 available from NSIDC, covering the period 1975 - 1984. After delineation of fast ice areas it was possible to provide snow and sastrugi depth estimates on that type of ice.

Finally, attempts were made to look at contemporary snow depth data in order to compare with results from the Sever expeditions. We analyzed buoy snow depth data since those data has been collected in the area that overlaps with the space observed during Sever expeditions. It is difficult to compare contemporary and historical data because of the different methodologies in collecting data. However, it was important for us that besides providing in situ snow depth observations, buoy measurements demonstrate the range of temporal and spatial variability of snow depth in the end of winter.

# 4 Results

## 4.1 Depth of undisturbed snow cover

The spatial analysis of snow depth changes from the Sever data is based on the observations made on the prevailing ice types in the landing areas, consisting of level FY or MY ice. Since the landing sites were irregularly distributed, the basic statistical characteristics were calculated in two ways: (1) from point measurements and (2) from gridded data as shown in Fig 5b. The statistics for the two methods is presented in Table 2. The snow in the form of sastrugi, attached to hummocks and ice ridges is not described here; the relevant analysis will be presented in Sect. 4.2 and 4.3. Data from the Barents Sea is not included because of very few measurements in that region.

**Table 2**. Snow depth ($H_s$) of undisturbed snow cover on level ice in the landing areas in different parts of the Arctic Ocean for the months March, April and May.

| Snow depth (cm) | Central Arctic[*] | Kara Sea | Laptev Sea | East-Siberian Sea | Chukchi Sea |
|---|---|---|---|---|---|
| Statistics based on the gridded data | | | | | |
| **Average** | 21.0 | 10.3 | 9.8 | 15.3 | 13.3 |
| **Std** | 5.5 | 3.6 | 3.1 | 3.4 | 3.1 |
| **Number of measurements** | 460 | 641 | 442 | 368 | 227 |
| Statistics based on the point data | | | | | |
| **Average** | 21.2 | 12.2 | 9.6 | 15.2 | 13.5 |
| **Std** | 10.9 | 12.2 | 8.4 | 10.1 | 10.5 |
| **Median** | 20 | 8 | 7 | 15 | 10 |
| **Min** | 2 | 1 | 1 | 1 | 2 |
| **Max** | 70 | 97 | 65 | 60 | 60 |

*) All data including the number of measurements correspond to observations carried out on the MY ice (ice with the thickness >200 cm).

Average snow depth estimated from point measurements and gridded data is similar in all regions except in the Kara Sea. The difference of about 2 cm in this region can be explained by very selective sampling in 1960s, when most of measurements were carried out in the north-eastern part of the Kara Sea near the coast and in the Vilkitsky Strait. The reason for this biased sampling is not known. The maximum average snow depth of about 21 cm was observed in the central Arctic. Among the marginal seas, the East-Siberian Sea showed the largest fraction of snow depths above 15 cm, as shown in distributions (Fig 6).

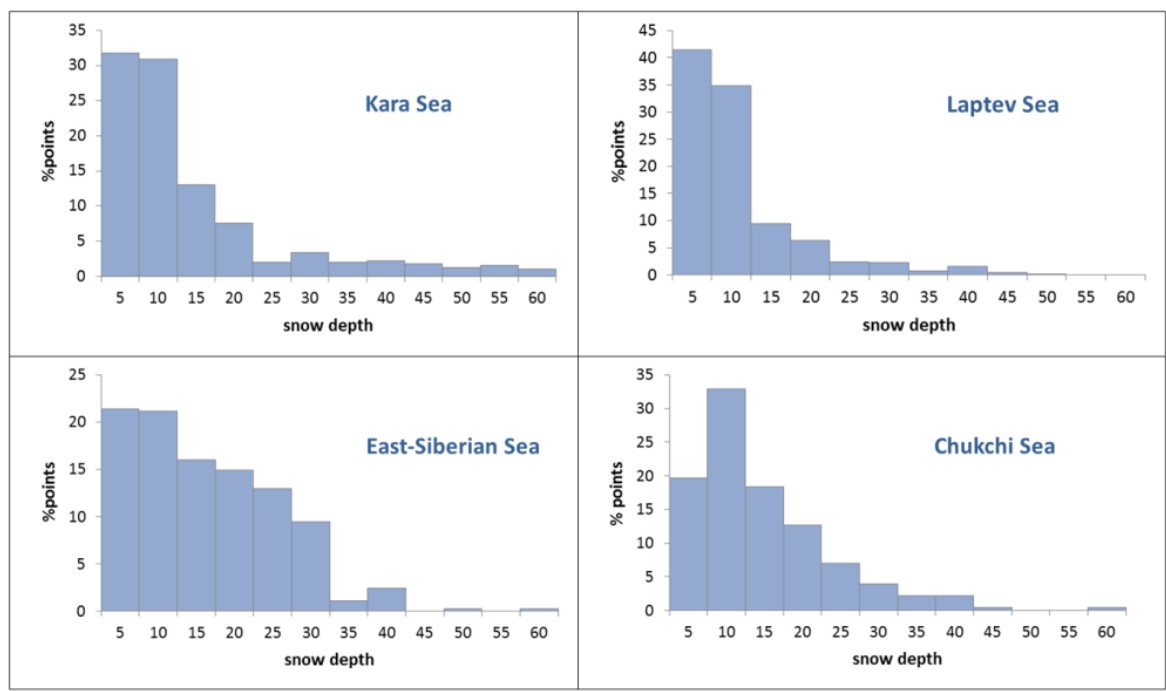

**Figure 6**. Distribution of snow depth on the level ice in the marginal seas.

The snow depth in the Kara and Laptev seas is dominated by snow depth up to 10 cm, indicating prevalence of FY ice. The Kara Sea also has cases of very thick snow cover, which can be explained by high density of data in the north-eastern part of the Kara Sea as shown in Fig 5. Snow conditions observed in that area were highly variable, which is explained by very condensed observations that captured a wide variety of ice and snow characteristics in that region. On average, the East-Siberian and Chukchi seas had more observations of thicker snow cover, which agrees with the higher proportion of MY ice in these regions.

The start time of snow accumulation is one of the major factors determining the snow depth by the end of snow accumulation season (Radionov et al., 1997; Hezel et al., 2012). In the case of FY ice, snow accumulation can only start after the sea ice freezing is stable. A delayed sea ice freeze-up will lead to a delayed start of snow accumulation and thereby have impact on the snow depth evolution during the winter (Webster et al., 2014). MY ice begins to accumulate snow earlier in the fall; additionally, in some regions the snow can survive through the summer season. The ice thickness generally increases throughout the cold season, and though the sea ice growth rate is known to be inversely proportional to its thickness (e.g., L'Heveder and  Houssais 2001, Bitz and Roe, 2004) we would expect  the ice thickness to be related to the depth of snow accumulated during the winter. Figure 7 plots the depth of snow versus the ice thickness. The data are averages corresponded to ice thicknesses divided into 20 cm-ice-thickness intervals. All measurements used here were conducted throughout the Arctic (see Fig. 5a) in the MAM months on the undeformed ice.  The number of measurements is shown by the histogram.

An empirical linear relation between ice thickness and snow depth can be derived for the FY ice, using least square regression, as shown in Fig. 7 by the red line. The derived relation is the following:

$$H_s = 0.069 * H_i + 2.0. \tag{1}$$

In the equation $H_s$ is the snow depth of the undisturbed snow on the undeformed ice and $H_i$ is the ice thickness (both in cm).

5   The FY ice has been separated using a threshold of 200 cm. The linear regression was carried out using averaged snow depth and mean ice thickness for each 20 cm-ice-thickness group of data. The coefficient of determination ($R^2$) is 0.95. The relation between snow depth and sea ice thickness for the whole data set can be described by the polynomial function (see Fig. 7). The highest values of the ice thickness from the lowest number of measurements were rejected from the polynomial calculation.

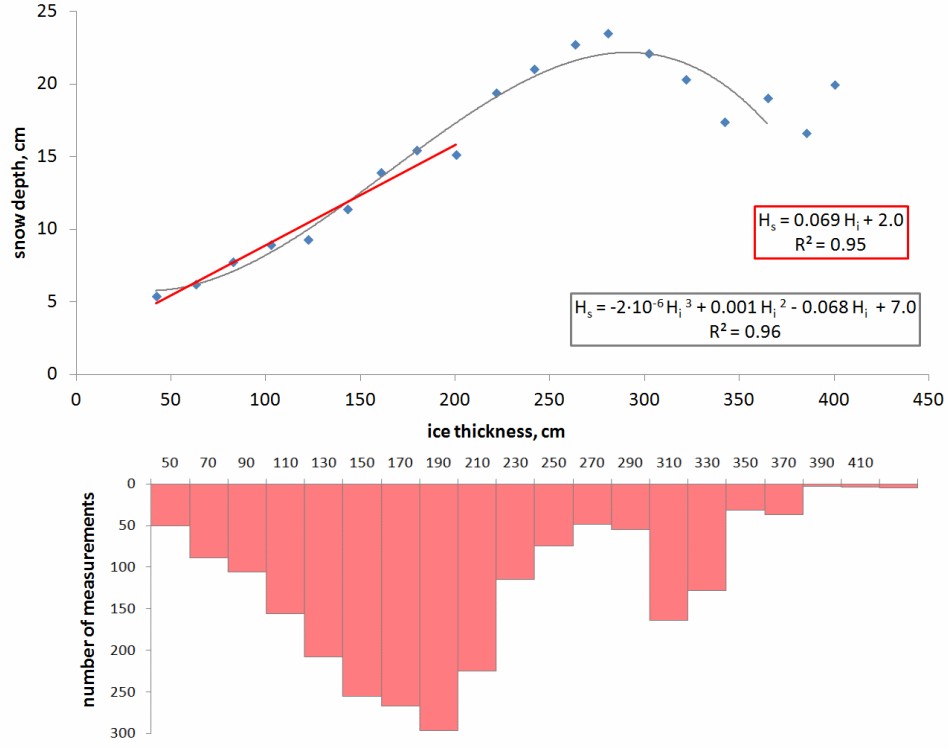

**Figure 7**. The relation between the thickness of the undeformed ice and the depth of accumulated snow in the end of winter.
The next step in the analysis of the Sever data was focused on the local variations of snow depth caused by wind action and ice deformation, leading to formation of sastrugi, hummocks and snow dunes. As described in Sect. 2, measurements of snow depth of sastrugi (sastrugi height), snow depth on hummocks and depth of snow dunes connected to ice ridges have

15   been done at a significant number of landings (Fig. 3). That data allows us to evaluate local variations of snow depth on single floes and to analyze spatial variability of snow irregularities over larger areas.

## 4.2 Snow depth of sastrugi

Observations of sastrugi height started in 1963, resulting in measurements from 1748 landings. Sastrugi area observations started in 1974, providing data from 1217 landings. A map of measured sastrugi heights is shown in Fig. 8a. Similar to the case of snow depth measurements, the highest density of observations was in the marginal seas.

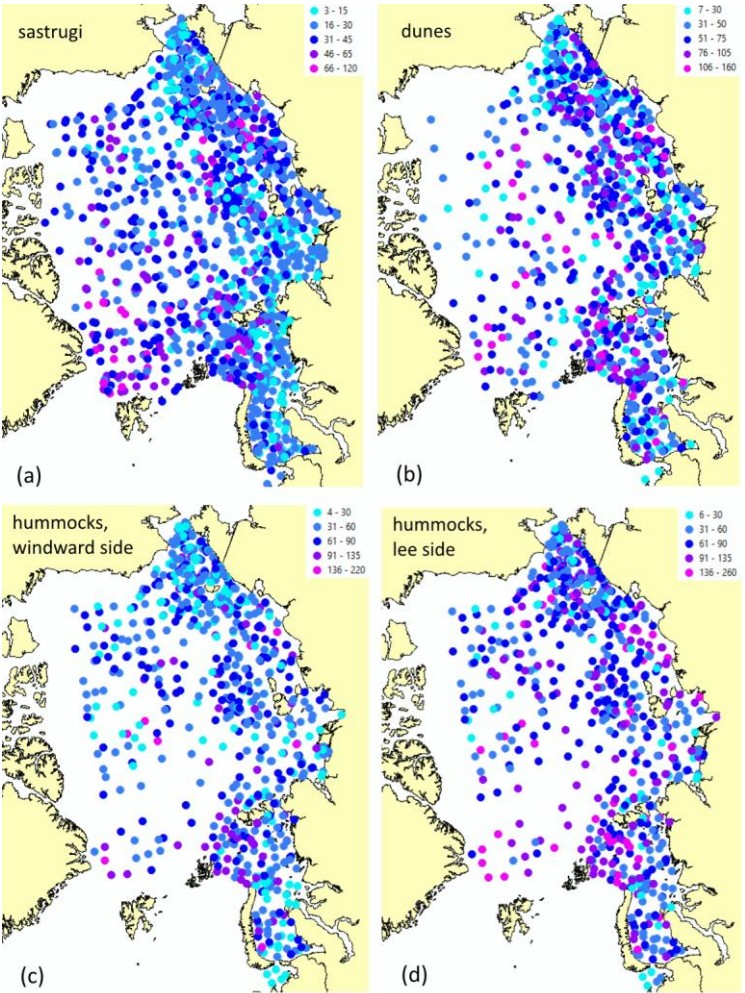

Figure 8. Height of sastrugi (a), depth of snow dunes extending out from ice ridges measured at a mid-length of a dune (b), snow depth on hummocks measured on the windward side (c) and snow depth on hummocks measured on the lee side (d), in cm. All observations were made in the MAM months.

It is reasonable to assume that the sastrugi height ($H_{sas}$) is related to the depth of undisturbed snow ($H_s$) because both parameters were obtained from the same landing areas. To calculate such empirical relation snow data collected in the MAM months were grouped into 5 cm-snow-depth intervals, and the regression between averages in each interval were computed resulting in the following equation: $H_{sas} = H_s + 15.5$ with the coefficient of determination ($R^2$) of 0.99, where $H_{sas}$ and $H_s$ are

in cm. The average sastrugi height variation of 10 cm is nearly constant over the full range of measured $H_s$ and $H_{sas}$ values. This implies that in the case of small snow depth the height of sastrugi can be ten or even more times higher than the surrounding snow depth.

The relation between level snow depth and sastrugi height is used later in the study to estimate average snow depth on the ice surface in order to make Sever expeditions data compatible with NP stations data. Since area covered by sastrugi is only a part of the ice surface, it is important to assess the extent of sastrugi areas, which started to be observed from 1974.

Sastrugi height and area of FY and MY ice has been analyzed separately to get statistics for the two different ice types. The average sastrugi height on the MY ice was 35.4±16.0 cm and the average sastrugi area was 29.3±18.1% of the ice surface observed during the Sever expeditions. By comparing monthly mean values for the MAM months, the height of sastrugi on the MY ice had increased from 31 cm in March to about 38 cm in May. The average sastrugi height on the FY ice varied from about 18 cm on the thinnest visited ice to about 28 cm on the thickest FY ice (with the ice thickness from 190 to 200 cm). It is important to note that there was relatively little number of measurements on the thin ice. The sastrugi area changed from about 31% on the thin ice to about 47% on the thick ice. The monthly averaged sastrugi height on the FY ice increased from about 22 cm in March to about 31 cm in May.

**Table 3**. Summary statistics of the height and area of sastrugi on the prevailing type of ice in the landing area in the Central Arctic and in the marginal seas of the Arctic Ocean.

| Statistics | Central Arctic | Kara Sea | Laptev Sea | East-Siberian Sea | Chukchi Sea |
|---|---|---|---|---|---|
| Sastrugi, cm | | | | | |
| Average | 36.2 | 27.0 | 23.0 | 31.8 | 25.4 |
| Std | 15.8 | 15.3 | 11.9 | 15.8 | 12.8 |
| Median | 35.0 | 25 | 20 | 30 | 25 |
| Min | 5 | 4 | 3 | 5 | 3 |
| Max | 120 | 93 | 85 | 100 | 90 |
| Number of measurements | 489 | 458 | 293 | 290 | 218 |
| Area of sastrugi, percent of the ice area | | | | | |
| Average | 29.3 | 34.9 | 38.1 | 40.7 | 32.2 |
| Std | 18.1 | 20.3 | 24.6 | 20.6 | 20.8 |
| Median | 20.0 | 30 | 30 | 40 | 30 |
| Min | 2 | 3 | 4 | 2 | 3 |
| Max | 85 | 100 | 100 | 90 | 90 |
| Number of measurements | 285 | 293 | 226 | 242 | 171 |

The sastrugi height in the marginal seas is the result of the length of snow accumulation period, wind activity, type of ice surface and some other factors. The summary statistics of the height and area of sastrugi for the marginal seas are presented in Table 3, showing that the sastrugi properties follow the level snow depth in different seas (shown in Table 2). For example, the largest sastrugi height is found in the East Siberian Sea which also has the largest level snow depth. In the Kara Sea relatively high value of the average sastrugi height is followed by high standard deviation. This may be explained by the high number of measurements in that region. Variability of the estimated sastrugi area is from 50 to 64% of the reported value that is predominantly governed by the natural diversity of the parameter.

Attempts to derive a relation between sastrugi area and sastrugi height did not provide any result in the case of MY ice. Generally, there is an increase of sastrugi area with increasing height; however the variability of the area values is very high. In the case of FY ice, the regression between $H_{sas}$ averages in 5 cm intervals and corresponding sastrugi area values results in a sigmoid curve (an "S"-shaped curve) with the minimum value of about 25 cm, maximum of about 45 cm. The variability of the area was as large as 88% at the lower $H_{sas}$ values end and 35% at the highest $H_{sas}$ values end.

### 4.3 Depth of snow on hummocks and depth of snow dunes

Snow depth on hummocks was measured at more than 1300 landings in the period from 1972-1986 (Fig. 8, c and d). The estimated area covered by hummocks was 24% of the observed ice surface where hummocks existed. The snow depth measured on the windward surface of a hummock was typically 23 % smaller than magnitude observed on the leeward side, however in 25% cases equal depths of snow were measured on both sides of the hummock. The average snow depth was 58.8 ± 35.3 cm on the windward side of the hummock and 76.6 ± 36.5 cm on the lee side. The maximum snow depth observed on hummocks was found in the East-Siberian Sea in 1982 when 220 cm was measured on the windward side and 260 cm on the lee side. Average values of the hummock snow depth are highest in the central Arctic (Table 4) that can be expected. Differences in average snow depths on hummocks in the Siberian seas can be the result of selectiveness of measurements The histogram of snow depth around hummocks for the central Arctic is the smoothest in comparison to marginal seas (Fig. 9). Asymmetry in distributions of the snow depth on hummocks in the seas may be the result of stronger and changeable winds and also perhaps a consequence of the movement of the floes. For example, rotation of the floes can swap windward and leeward sides of the hummock.

Depth of snow dunes extending out from ice ridges has been measured at mid-length at 1012 landings from 1974 – 1986 (Fig. 8b). The average observed value was 57.1 ± 27.0 cm and the maximum was 160 cm. Values obtained in the central Arctic were 16% higher than in the marginal seas (Table 4). Difference in the average values for marginal seas is perhaps the result of the selectiveness of measurements.

From the Sever observations, the snow depth of the dunes behind ice ridges could be estimated in relation to surrounding snow cover. Note that this characteristic cannot be obtained through comparison of average values of snow depth on the level ice (Table 2) and depth of the dunes behind ice ridges (Table 4) because the depth of snow dunes was not measured at every landing. The average (over the whole Arctic) snow depth on the level ice in the neighborhood of ice ridges was 13.8

cm with the median of 10 cm, being lower than average from all landings. It means probably that if there is a trap for blowing snow like an ice ridge, snow is relocated there from the level ice into considerable accumulations. Comparison of the depth of snow dunes  extending out from ice ridges and snow depth on the surrounded ice measured during the same landing showed that in the central Arctic the depth of dunes measured on their mid-length was 4.5 times greater than the level snow cover. In the marginal seas the dunes in the midpoint of their length were 4.5 - 8 times higher than the surrounded snow cover.

The depth of snow dunes that stretch out from ice ridges depends on the height of the ridges. Sea ice ridges are elevated structures that form when ice floes are moved by wind, ocean currents, or other forces relative to each other that results in colliding and producing a lot of ice fragments that are piled up along a line, with the steep-sloped edge. The height of the ridge above sea level depends on several factors, the thickness of the compressed ice being one of them. Hibler et al. (1972) provided the value of about 1.3 m as an average sail (a part of the ridge that is above the water surface) height basing on data collected in the Baffin Bay in 1970 on the ice of 1.2 m and 0.6 m average thickness. During Sever expeditions a parameter called "prevailing height of ridge hummocks" was observed. In the description of the ice measurements during a landing it is indicated that "the heights of several typical ridges were measured, with 5-10 measurements on each ridge", thus we can use the mentioned parameter as a reasonable representation of the average sail height.  The data have been collected over the whole Arctic area and the average sail height measured on the ice with the thickness less than 2 m (1504 landings) was about 1.5 m and the same average on the thicker ice (818 landings) was about 2 m. We can assume that the most heavily ridged ice was inaccessible for airborne expeditions and reported values most probably underestimate the real height of ridges.

The distribution of depth of snow dunes extending out from ice ridges (Fig. 9) reflects differences in morphology of sea ice in different parts of the Arctic, and partly, probably, is the results of bias in the sampling of measurements.  In the central Arctic the highest values were observed in connection to the largest ice ridges (Table 4). MY ice is exposed to various deformations in different years and structure of a ridge undergoes continuous evolution due to new ridging, freezing, melting and erosion. MY ice ridges therefore become smoother with time; along with that fresh ridges emerge and are present among the old ones.

In the Kara Sea, there was the highest number of measurements comparing to other seas that probably allowed observing most cases of snow dune layouts and states. The ice where measurements were conducted in that sea was quite thin, with the average thickness of about 110 cm; however the prevailing height of ridges averaged over all landings was 153 cm, being close to the same parameter in the East-Siberian and Chukchi seas (159 cm) where presence of MY ice implies existence of higher ridges. Availability of high snow dune depths (see the tail in the histogram, Fig. 9) that were observed near large ice ridges provides a high average snow dune depth of 57.0 cm.  In the Kara Sea the difference between the depth of snow dunes and the depth of snow cover in the landing area was the highest, being 48.7 cm on the average. In the Laptev Sea, the prevailing height of ice ridges was estimated at 140 cm and averaged depth of snow dunes attached to ridges was 51.2  cm. The smoothness of the histogram (Fig. 9) is perhaps a result of equal representation of all heights of ridges that happened to exist on the landing spots. More irregular histograms for the East-Siberian and Chukchi seas (Fig. 9) is caused by presence a

substantial fraction of the MY ice there. Average depth of snow dunes in the East-Siberian Sea was the highest among all marginal seas, being 59.0 cm on the average. Lack of the highest values of the snow dune depth in the Chukchi Sea can be explained by strong winds that are typical for that area (Martin et al., 2014) or perhaps by insufficiency of measurements. Collected observations resulted in 49.5 cm average snow depth of the dunes attached to ice ridges in the Chukchi Sea, with

the lowest difference between the depth of dunes and the depth of snow cover in the landing area (36.1 cm).

**Table 4**. Depth of snow in dunes extending out from ice ridges and on the hummocks in different parts of the Arctic.

| Region | snow dunes | | snow on hummocks | |
|---|---|---|---|---|
| | average depth, cm | number of measurements | average depth windward/lee side, cm | number of measurements |
| Central Arctic | 64.9±28.9 | 241 | 64.4±29.7 / 81.8±36.3 | 171 |
| Kara | 57.0±28.0 | 222 | 59.8±31.2 / 78.3±38.2 | 136 |
| Laptev | 51.2±26.4 | 157 | 55.2±28.2 / 71.5±36.9 | 95 |
| East Siberian | 59.0±25.3 | 215 | 57.6±30.1 / 76.2±36.6 | 142 |
| Chukchi | 49.5±21.2 | 161 | 50.4±26.1 / 72.3±34.9 | 172 |

## 4.4 Snow depth on fast ice

Fast ice is the part of sea ice that stays relatively immobile because it is attached to the coastline or to the shallow sea floor. The most extensive fast ice cover is formed in the Laptev, East-Siberian and Kara seas; the East-Siberian Sea is

characterized by the largest fast ice area in the Arctic and the greatest interannual variability (Johannessen et al., 2007, Yu et al., 2013). Since fast ice in general does not move, it provides conditions where snow can accumulate from the beginning of winter being relocate only by the wind. Fast ice is smooth when it forms but can be deformed during wind events or when drifting ice is pushed against the seaward boundary of the fast ice. In these cases the ice can become highly deformed with shear ridges or stamukhi (Barry et al., 1979). The areas of stamukhi were not observed by Sever expeditions because

landing was not possible.

The average snow depth on the prevailing ice of landing area on the fast ice was 15 ± 13 cm with the average ice thickness where landings occurred being 169 ± 48 cm. The average sastrugi height was 29 ± 16 cm. How average snow depth and sastrugi height change from one area to another is illustrated by Table 5. There is also indication of the proportion of measurements carried out on the fast ice in comparison to all measurements performed in the sea.

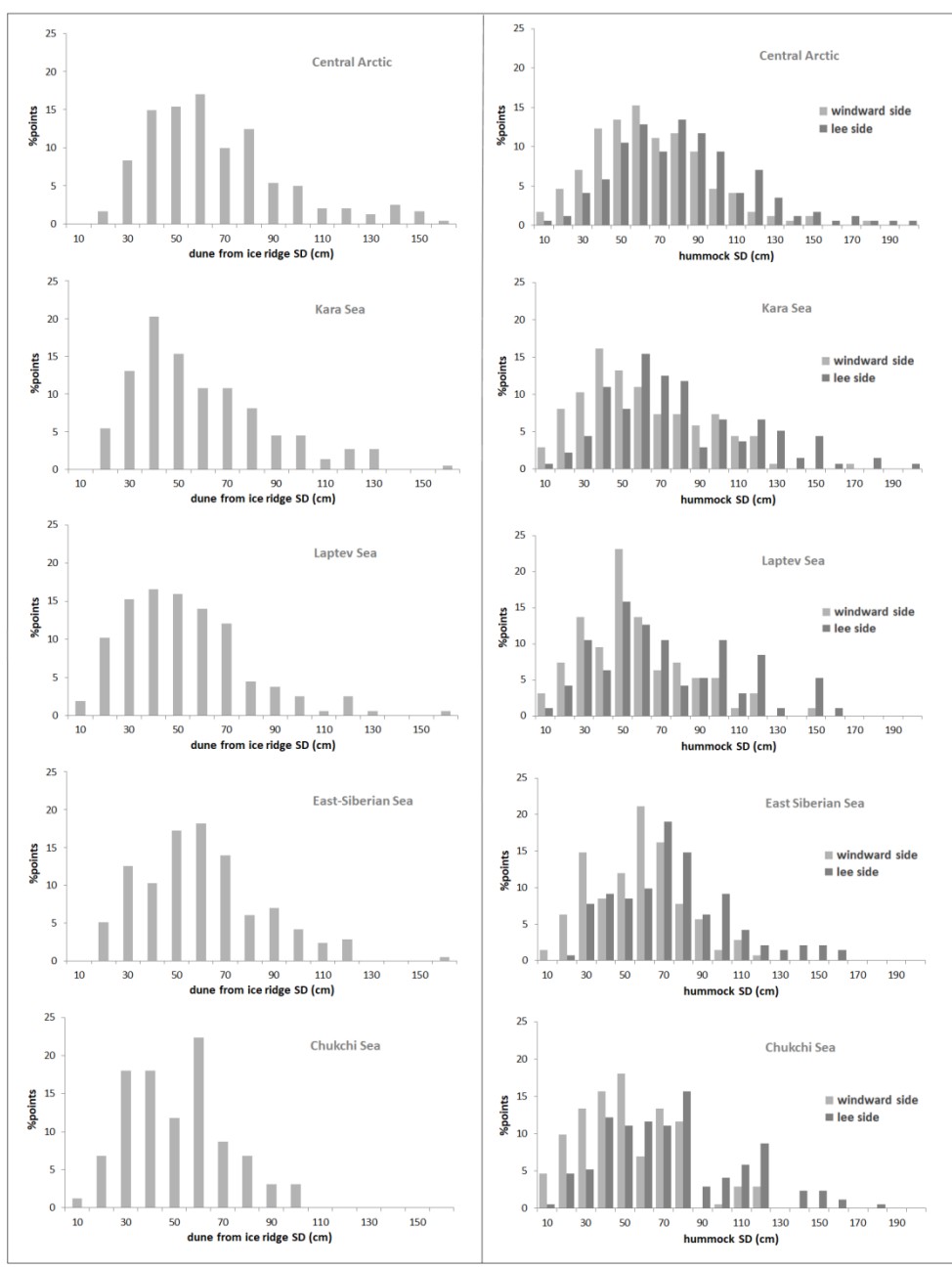

**Figure 9**. Distribution functions of snow depth (SD) in the dunes attached to ice ridges and around hummocks (on the windward and on the lee side) in different parts of the Arctic.

The small number of measurements in the Chukchi Sea was caused by limited possibilities of taking such observations because the fast ice in this region is formed in a narrow zone near the coast. The other marginal seas had much more landings on the fast ice. The analyzed data can be biased due to sampling, for example in the Kara Sea in the north-eastern part of the sea and in the Vilkitsky Strait, where sampling in the 1960s was much denser than in any of the other areas. There is lack of other studies that makes it difficult to assess the values shown in Table 5. The deepest snow covering fast ice was observed in the Kara Sea with the highest variability; the same is valid for sastrugi. That perhaps is (at least partly) a result of combined effect of the intensity of precipitation and air masses movement, which has specific characteristics because of existence of relatively sheltered areas. The lowest values of snow depth have been recorded in the Laptev Sea, which is characterized by comparatively low level of ridging (Eicken et al., 2005) allowing the snow to be blown off the ice surface.

**Table 5.** Average snow depth on the level ice and height of sastrugi in the fast ice area.

| Region | Snow depth on prevailing ice | | | Sastrugi | | |
|---|---|---|---|---|---|---|
| | Average depth, cm | Number of measurements | Percent of all measurements, % | Average height, cm | Number of measurements | Percent of all measurements, % |
| Kara | 17.0 ± 16.0 | 243 | 38 | 34.2 ± 18.4 | 181 | 39 |
| Laptev | 11.6 ± 9.1 | 204 | 46 | 22.2 ± 11.7 | 147 | 50 |
| East Siberian | 14.2 ± 8.4 | 151 | 41 | 32.9 ± 13.9 | 90 | 31 |
| Chukchi | 15.9 ± 13.5 | 15 | 7 | 23.3 ± 10.7 | 11 | 5 |

In the Kara and Laptev seas the average snow depth measured on the level fast ice (see Table 5) is higher than the snow depth on the level ice averaged for the whole sea (see Table 2). That can be explained by longer (on the average) period of ice existence in the case of fast ice. The standard deviation is also proportionally higher. In the East Siberian Sea the average snow depth on the fast ice is lower than the same parameter of the whole sea, probably because the relatively high proportion of MY ice outweighed fast ice in benefits for snow accumulation. In the Chukchi Sea the similar comparison is hardly have meaning because the number of measurements on the fast ice was too small.

**4.5 Combining NP and Sever data for the MAM months**

When merging NP and Sever data it is important to treat the data in such a way that comparison makes sense and a combined product is meaningful.

NP drifting station snow observations were not accompanied by any ice observations. Therefore the variability in NP snow depth measurements cannot be explained from different ice characteristics. That complicates processing Sever data with the aim to make them similar to NP data. The only information about NP snow measurements is that the NP data were collected from a solid MY floe. Figure 10 represents snow line observations made by NP16 drifting station personnel in 1969 in the

MAM months. It shows that spatial variability of measured values was very high. The range of values over the same line was from 2 to 64 cm in March, from 4 to 90 cm in April and from 5 to 90 cm in May. The low correlation between repeated measurement lines (0.18 between March and April and 0.04 between April and March) suggests that the variations were primarily caused by wind. Average snow depth increased from 26.4 cm in March to 40.6 cm in May and the number of high snow depths has increased correspondingly (Fig. 10 b-d).

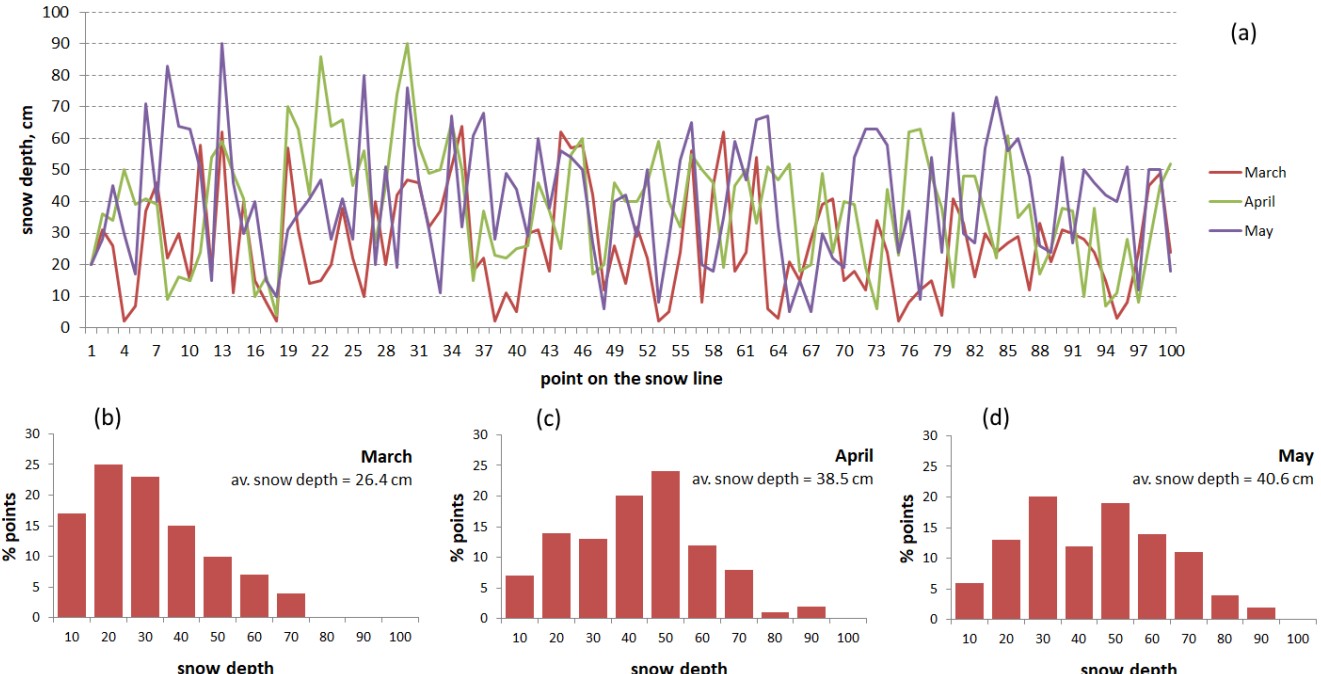

**Figure 10**. (a) Snow line measurements made in 1969 in the MAM months during the work of NP16 drifting station. The length of snow line was 1 km. (b) - (d) Distributions of measured snow depths in March, April and May 1969.

In order to produce an integrated map that describe the average state of snow cover in the MAM months, the Sever and NP measurements have been combined by gridding all Sever data on snow depth on the level ice ($H_{sev}$) together with NP data. The height of sastrugi weighted in proportion to the sastrugi area was added to each Sever snow depth in a grid cell using the formula

$$H_{sev} = H_s + P_{sas}^{reg} \cdot H_{sas} \; , \tag{2}$$

where $H_s$ is the depth of snow measured on the level ice (described by data providers as the snow depth on the prevailing ice in the landing area), $H_{sas}$ is the height of sastrugi, and $P_{sas}^{reg}$ is the average portion of the ice surface covered by sastrugi in the region of snow measurements (see Table 3). In the central Arctic, where the ice was mainly perennial at the time of measurements, only observations made on the MY ice during Sever expeditions have been included into computation as was explained in Sect. 3. The height of snow attached to ice ridges was not included in the calculations because 1) the Sever measurements in the Western part of the Arctic Ocean are too scarce, and 2) there are no estimations of the area covered by

such features from the Sever expeditions. The effect of not including that data results in some underestimation of the average snow depth. The SHEBA observations indicated that in April and May 1998 about 3.9% of the examined area was covered by deep snow (>80cm) associated with ice ridges (Sturm et al., 2002). The underestimation is most important in the western Arctic, especially north of the Canadian Archipelago, where the highest concentration of the ridged ice is expected (Bourke and McLaren, 1992, Makshtas et al., 2003, Shoutilin et al., 2005).

The snow depth distribution on the sea ice in the MAM months in 1960-80s has been generated using 2302 points, 143 of which were monthly averaged NP data and others were Sever snow depths calculated as described in the previous paragraph. Two approaches have been used to produce a new snow depth map for the MAM months in the Arctic. The first approach was based on gridding as described in Sect. 3 with a grid resolution of 100x100 km. The result is presented in Fig. 11a. The projection is North-Pole Stereographic with the central meridian 20° E and the latitude of origin 90°. Standard deviation of the data shown in Fig. 11a is represented by Fig. 11b. It is calculated as a weighted standard deviation from variances of $H_s$ and $H_{sas}$ and a weight of 0.35, which is an average portion of sastrugi area in the Arctic (see Table 3). The second approach was similar to the one used by W99. The two-dimensional quadratic fit has been calculated using the combined data set of 2302 measurements collected in March, April and May. The result is shown in Fig. 11c. It is a much smoother data set compared to map in Fig. 11a. The detailed mapping of the snow depth in the marginal seas is lost. The two-dimensional quadratic fit has been calculated as

$$H_s = H_0 + C_1 \cdot x + C_2 \cdot y + C_3 \cdot x^2 + C_4 \cdot xy + C_5 \cdot y^2 \ , \tag{3}$$

where $H_0$ = 35.05 cm, $C_1$ = -4.69$\cdot$10$^{-6}$, $C_2$ = -1.46$\cdot$10$^{-6}$, $C_3$ = -2.27$\cdot$10$^{-12}$, $C_4$ = 2.91$\cdot$10$^{-12}$, $C_5$ = -1.16 $\cdot$ 10$^{-12}$, x and y are coordinates in the North-Pole Stereographic projection. Units of x and y are meters. To support the comparison between the new and the W99 climatologies, Fig. 11d shows W99 climatology for the MAM months, which is an average of the mean snow depths for March, April and May.

The contour lines generated from the gridded data are shown in Fig. 11a where the spatial variability in the marginal seas is well captured. The lowest snow depth is found in the middle of Kara and Laptev seas where the sea ice was absent in the summer for most of the years. Increased values of snow depth near Novaya Zemlya, Severnaya Zemlya and in the Yana Bay is associated with the ice massifs in these areas consisting of thick rough ice. Relatively high values of snow depth in the East-Siberian Sea are connected to large amount of MY ice in the area. In the Chukchi Sea, there were less MY ice than in the adjacent East-Siberian Sea but more than in Laptev and Kara seas. In the central Arctic, there is much less data available for each grid cell, leading to larger uncertainty compared to the marginal seas. In the Canadian sector there is very little observation data and the map in Fig. 11a shows too little snow depth in an area where ice thickness is known to be largest.

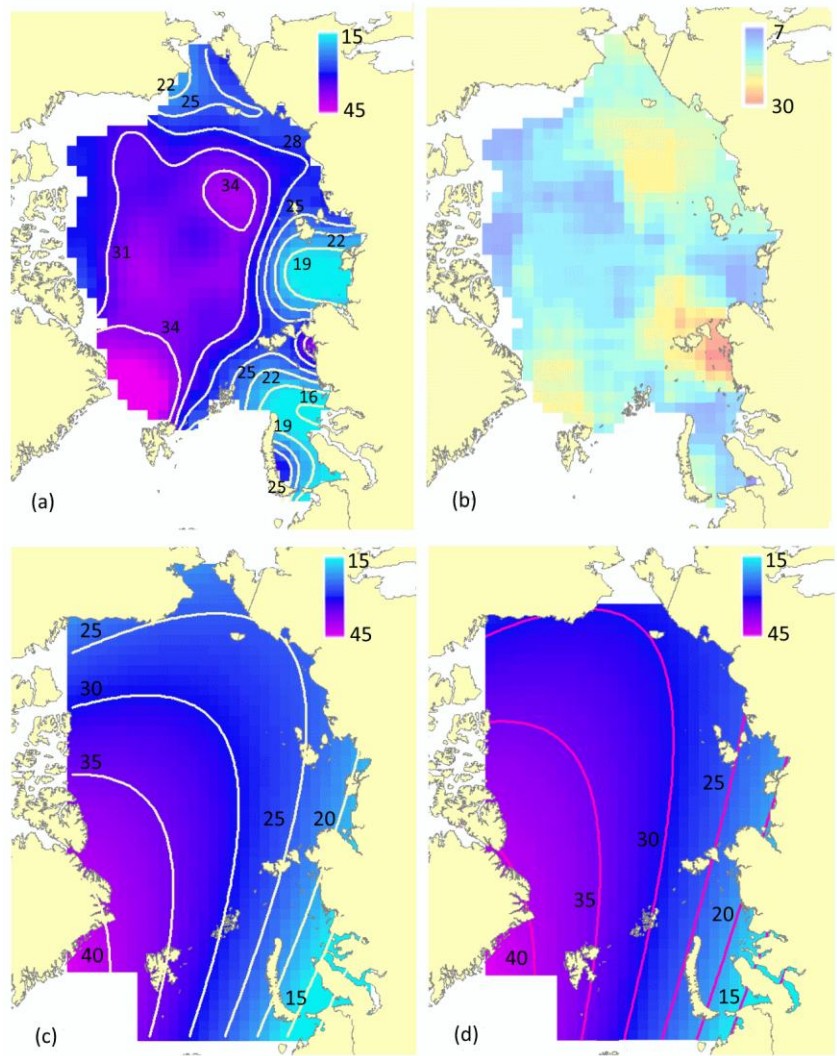

**Figure 11**. Map of snow cover depth (in cm) on the sea ice in the MAM months in 1959-1988: a) gridded data with the grid cell 100x100 km and contour lines overlaid on it, b) standard deviation of the data used for gridding, c) map produced from the same data as (a), but using two-dimensional quadratic fit shown by formula (3), d) Warren climatology averaged for the MAM months.

## 4.6 Contemporary in situ snow depth measurements

In the changing Arctic climate it is important to have contemporary data on snow parameters that can be compared to historical data such as the Sever data. After 1989 no extensive observations like those from the Sever expeditions have been collected. So there is no similar data from present years that can be compared with historical data to assess changes in snow cover on sea ice. In the last two decades in situ measurements of snow depth from various automated buoys have been

collected, in particular from Ice Mass Balance buoys (IMB), see (Richter-Menge et al., 2006, Polashenski et al., 2011, Perovich et al., 2013.). Buoy measurements provide time series of the snow depth gained from the same position on an floe, providing good data on time evolution of snow depth, but very limited data on spatial variability. The Sever data were opposite providing good spatial coverage, but very limited temporal coverage. The NP data were more similar to IMB data,

but allowed spatial sampling inside a 1 km grid. Attempts were made to compare the Sever/NP estimates with contemporary IMB estimates in order to detect possible trends in snow depth over the last five – six decades.

Snow depth data from selected IMB buoys that were deployed on sea ice and operated through the winter season from 2011 to 2015 have been analyzed for the MAM months. The buoys, which were provided by CRREL, were placed on the level ice, measuring snow depths at specific sites of MY ice floes. Changes in snow depth were therefore caused by precipitation

and wind action. Time series of snow depth from buoys in different parts of the Arctic are presented in Fig. 12. The snow depth variations over the three month period are small, except for a few buoys which registered significant changes. We should note that a ~25 cm increase of snow depth within a very short time period registered by the buoy 2013F seems unrealistic. The average snow depth from all the buoys in the MAM months is 27 cm while the median depth is 24 cm. The time-averaged snow depth for each buoy varies from less than 10 cm (2011J) to more than 50 cm (2013F). The spatial

variability in snow depth from the 10 buoys seems to be random. The average IMB snow depth of 27 cm compares well with the average Sever snow depth of 21 cm in the central Arctic, but different sampling schemes makes it difficult to draw any conclusion from these measurements.

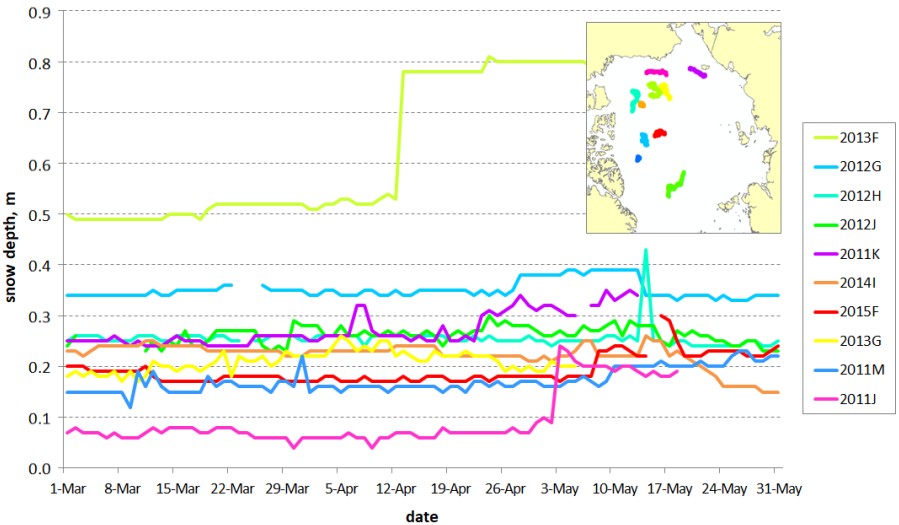

**Figure 12**. IMB observations of snow depth in the MAM months, 2012-2016, provided by CRREL. The number before a

letter in the buoy's name indicates the year when the buoy has been deployed. Tracks of the buoys in the MAM months are shown on the map.

Another snow depth data set is provided by AWI (Nicolaus et al., 2016), which deployed snow buoys on FY ice floes in the central Arctic in 2015. The FY floes became second-year ice in the following winter. The buoys drifted towards the Fram

Strait as shown in Fig. 13. The average snow depth in the MAM months of 2016 was 21±9 cm, which is very similar to the Sever data in the central Arctic. Although the buoys were located quite close to each other they measured rather different snow depths, varying in the range from 10 to 40 cm. This is another example of large spatial variability in the snow depth within a relatively limited area, illustrating the challenge of sampling snow depth on Arctic sea ice.

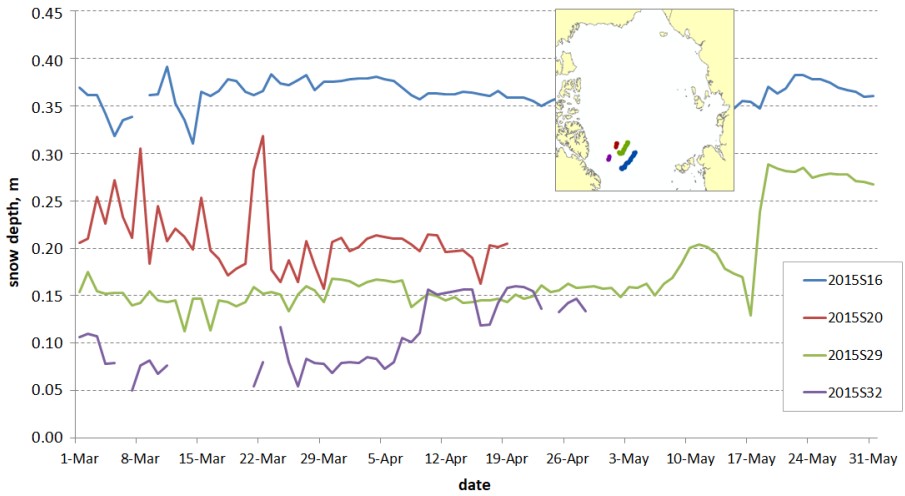

**Figure 13**. Snow depth AWI buoy in the MAM months of 2016. All buoys measured snow depth on the second-year ice. The plots show daily averages of snow depth from four buoys.

## 5   Discussion and Conclusions

This paper presents an analysis of snow depth and other Arctic sea ice data from the Sever expeditions and the North Pole
drifting stations in the period 1959-1989. By merging snow depth data from the two extensive observing programmes, a new snow depth climate data set has been provided for the end of winter season (March, April and May). This data set is an important extension of the previous snow depth climatology provided by W99. The data set is also important as a reference data set for comparison with contemporary and future observations of snow cover in the Arctic.

The Sever measurements were obtained over large parts of the Arctic sea ice mainly during March, April and May when the
aircraft could land on sea ice. Snow and sea ice data were collected both from the runway and the surroundings of the runway, called "prevailing ice of the landing area". A total of 3234 landings were conducted, of which 2331 provided snow depth measurements. The landing areas were chosen in order to collect representative data in different parts of the Arctic Ocean. Analysis of the Sever data for the whole Arctic shows that the average snow depth of level ice was 14.3 cm with the standard deviation 11.9 cm. The average depth of sastrugi was 30.1±15.7 cm, of snow dunes attached to ice ridges
57.1±27.0 cm, and of snow on hummocks 58.8±35.3 cm (windward side) and 76.6±36.5 cm (lee side).

The results for different Arctic regions are presented in Tables 2, 3, and 4 and summarized in Fig. 14. The Central Arctic had an average snow depth on level ice of 21 cm, while it varied from 9.6 to 15.2 cm in the marginal seas. The average sastrugi

depth varied from 23.0 to 35.4 cm. The average depth of snow attached to ice ridges was in the range from 49.5 cm in the Chukchi Sea to 64.9 cm in the Central Arctic. On a windward side of the hummock the average snow depth varied from 64.4 cm in the Central Arctic to 50.4 cm in the Chukchi Sea. On the lee side the average snow depth was in the range from 71.5 cm (Laptev Sea) to 81.8 cm (Central Arctic). The average snow load on the ice in the Arctic can be described as a

combination of the depth of undisturbed snow on the level ice and snow depth of sastrugi that covered about 35% of the ice surface. Furthermore, the average snow depth should include a contribution from snow connected to hummocks and ridges. But since there was no adequate data on the area covered by hummocks and ridges from Sever expeditions, this factor could not be included in the snow depth climatology, implying that it is probably underestimated.

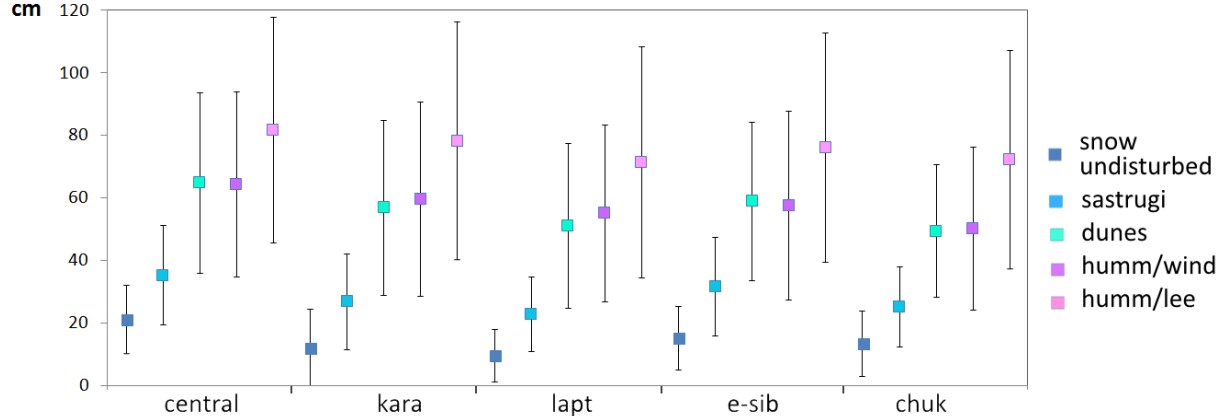

**Figure** 14. Sever snow depth observation statistics in the MAM months: depth of undisturbed snow on the level ice, snow depth of sastrugi, snow depth of dunes extending out from ice ridges, depth of snow on windward and leeward sides of the hummock and corresponding standard deviations. Data shown are averages for the central Arctic and for the four seas: Kara, Laptev, East-Siberian and Chukchi.

The snow depth data set by W99 was obtained by gridding of relatively limited data from the North Pole Drifting Stations
using a quadratic fit. The present data set, which has been obtained by averaging a much larger number of observations from the Sever expeditions, especially in the marginal seas, is therefore more representative than the W99 data for the MAM period. Furthermore, the new data set shows a lower average snow depth in the central Arctic compared to W99. That can be explained by the fact that the NP data was collected on one MY ice floe per year, while the Sever data aggregated observations from a large number ice floes, providing a better representation of the spatial variability. The NP data, on the
other hand, provided information on the seasonal evolution of the snow depth on a single MY-floe. The NP data could therefore be used to estimate the mean temporal standard deviation for the MAM period, which was 3.6 cm. A similar temporal standard deviation could be derived from contemporary buoy measurements, which was 3.7 cm for the 10 CRREL buoys and 3.1 cm for the four AWI buoys.

The W99 climatology is based on data collected on the MY ice mainly in the central Arctic and is therefore only valid for
snow on MY ice. In the marginal seas, there is a mixture of MY and FY ice, which have very different history of snow

accumulation. All the marginal seas are completely covered by ice in winter (with the exception of the south-western part of the Barents Sea). FY ice was dominant in most seas, fast ice was present along the coasts and MY ice was present in various degrees. The current snow depth climatology is based on 1699 landings in the marginal seas, 460 Sever landings in the central Arctic Ocean and 143 NP averages for the MAM months. The snow depth climatology from this study should therefore be well representative for the three decades when data were collected.

In the last decades, the Arctic sea ice has changed significantly, in particular due to the reduction of MY ice, increased fraction of FY ice, with a corresponding thinning of the total ice cover (e.g. Wadhams, 2012, Kwok and Cunningham, 2015, Tschudi et al., 2016). Later onset of freeze and correspondingly later start of snow accumulation is the last but not the least factor that determines snow depth distribution in present time (Wang et al., 2013, Stroeve et al., 2014, Webster et al., 2014). The snow depth climatology from the Sever data from the period 1959 – 1989 is therefore not necessarily valid for the present situation in the Arctic. But results of the study can enlighten several aspects of the snow on ice problem which is important today. The difference between MY and FY regarding snow cover has impact on estimation of energy fluxes from ocean to atmosphere in coupled models, on satellite altimeter retrievals of ice thickness and passive microwave retrievals of thin ice thickness (Tian-Kunze et al., 2014, Key et al., 2016).

Most in situ snow measurements that were taken during validation campaigns in the last 20 years were collected in the Western Arctic covering near-shore regions of the Chukchi Sea, Elson Lagoon, some areas in the Beaufort Sea, some places in the Canadian Arctic among the islands and in a near-shore area close to Greenland. Those regions are poorly represented in the Sever data collection or not represented at all. Besides, the mentioned observations describe snow conditions in a particular year when the measurements were conducted that can differ from the average. However, it can be noted that the mean snow depth of $33.7 \pm 19.3$ cm measured at SHEBA during April and May (Sturm et al., 2002) is comparable with the snow depth for that area in the present gridded data. In our case the value is a bit lower and it is most probably a consequence of not including snow attached to ice ridges into calculation together with low density of available Sever measurements in that area. Average snow depth reported from Navy Ice Camp located at 72°55' N, 147°34' W, closer to the coast in comparison to SHEBA, was $20.6 \pm 18.8$ cm (Sturm et al., 2006), which is also in a good agreement with the gridded Sever data. Observations in an area north of the Greenland coast describe average snow depth as $25.7 \pm 26.3$ cm on the combinations of FY and MY ice near the coast (Farrel et al., 2012) and as $41.5 \pm 19.6$ cm on the MY ice 50 km off the coast (King et al., 2015), and that range of values agrees with the new climatology. Snow depth measured during the N-ICE2015 campaign in the area north of Svalbard in the end of winter was $52 \pm 12$ cm over the second-year ice and $33 \pm 14$ cm over the FY ice (Gallet et al., 2017). It is higher than the average climatology suggests, however, along with that, it is within the range of observed snow depth and sastrugi height values in that region. The average snow depth on the MY ice in the central Arctic for the MAM period is $24.3 \pm 0.7$ cm according to recent IMB buoy measurements (from four buoys 2011M, 2012G, 2012J, 2015F, see Fig. 12) and $21.2 \pm 9.4$ cm according to AWI snow buoy measurements. These values are in agreement with the measurement of snow depth on the level MY ice in our study.

The existence of ice ridges and hummocks has significant impact on the snow distribution on sea ice. The Sever expeditions provide useful data to quantify properties of snow accumulated around these features. But ridges and hummocks are also changing over last decades due to less MY ice and more FY ice. According to Wadhams (2012), the reduction in ice thickness has been accompanied by loss of ice in ridges, particularly in the MY ice. For FY ice the average sail height has been reported to be about 0.7 m (Strub-Klein and Sudom, 2012) for some parts of the Arctic using data collected in the period from 1976 to 2011. This estimate can be compared with similar data from the Sever expeditions, showing an average sail height of 1.5 m.  This suggests that snow dunes attached to FY ice ridges have been reduced significantly.

The Sever expeditions represented a unique observing programme in the Arctic, which is not likely to be repeated anytime in the future. Present and future observations of snow and sea ice will rely on satellites, aircraft and automated buoys, as described in Sect. 4.6. Satellite observations from altimeters, Synthetic Aperture Radar and optical /infrared sensors will be the backbone of a monitoring system for the Polar Regions, especially for sea ice and snow measurements.  In addition, a network of ice buoys to observe temporal changes and regular aircraft/UAV surveys to observe spatial variability will be essential to monitor snow and other sea ice properties as supplement to and validation of the satellite measurements.

*Data availability*.

The data generated within the research is openly available as a Supplement

*Author contribution*. E Shalina performed the data analysis and interpretation of the results. S Sandven critically revised the work and gave important feedback for improvement. Both co-authors participated in writing the manuscript.

*Competing interests*. The authors declare that they have no conflict of interest.

*Acknowledgments.* We acknowledge the great effort done by the Sever Program and North Pole drifting station program to collect the unique snow data which have been used in this study. The snow data from Sever program was obtained from NSIDC (http://nsidc.org/data/G02140). In particular Dr. V. F. Radionov has been  helpful providing the snow depth data from NP station observations. The new snow data from 2011-2016 were obtained from AWI (http://www.meereisportal.de/en/seaicemonitoring/buoy-mapsdata/) and CRREL (http://imb-crrel-dartmouth.org/imb.crrel/buoysum.htm) buoys archives. This research was supported by ESA Climate Change Initiative - Sea ice project, contract no. 4000112229/I15/I-NB and European Commission through EuRuCAS: European-Russian Centre for cooperation in the Arctic and Sub-Arctic environment and climate research, grant agreement no. 295068.

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
