# Peer review of "Snow depth on Arctic sea ice from historical in situ data"

_The Cryosphere, 2017_

## Referee Comment (RC1) · S. Hendricks (Referee) · 19 Jan 2018

The study "Snow depth on Arctic sea ice from historical in situ data" evaluates observations of snow and sea ice surface properties from the Sever aircraft landings on Artic sea ice from the 60th through 80th of the 20st century. The data contains snow depth from surfaces representative of the area near the landing site as well as some information on the snow depth distribution, from which the authors deduce average snow depth. This information is used to construct a climatological snow depth map based on significantly improved observational density in the Russian part of the Arctic compared to the Warren climatology. However, this climatology is only valid for the month between March and May due to visibility constraints for the landings.

[Figure]

The necessity and value to document and utilize these extensive observational datasets from the past cannot be understated and the paper constitutes a valuable contribution to this effort. The paper is generally well written and adds a thorough analysis to the documentation of the methodology. There are however a few general minor points and specific comments where the analysis and the presentation of the results could be improved before publication:

1) The authors provide a detailed climatology of average snow conditions but without a magnitude of the snow depth variability. It would be important to have this information as a measure the uncertainty of the climatology. Of course, variability can only be estimated in areas with repeated observations, but might be possible with pooling data on the Russian shelves.

2) The authors also did not show the difference to the Warren climatology. The improvements on the Russian shelves are obvious, but it would be valuable to assess the impact of the localized nature of the NP observations compared to the regional coverage of the Sever program on the generation of climatologies in regions where the observations should be comparable.

3) As the authors state themselves, the comparison with modern data is difficult due to different methodology of point measurements and the surveys at landing sites. Therefore, the result are not very insightful, especially without a magnitude of interannual variability in the timeframe of the Sever program. I would therefore suggest reducing the space allocated in the manuscript for this comparison.

Specific comments:

- replace 'fastice' with 'fast ice' throughout the document

- P10L16: Specify section number that you mean with "later"

- P12L6ff: Did the authors exclude MY thicknesses in this regression, because these would not be "undeformed"?

- P16L23: Rephrase sentence: "In the Kara Sea, there was the second (?) after ..."

- P20L6: Please provide the formula

- P20L6: Would it not be necessary to include the snow dunes in the estimation of average snow depth? Is there not information (ridge density) to do this?

- P20L15: Please provide coefficients of the quadratic fit

- P22L17: The author correctly state that it is difficult to draw any conclusions from a direct comparison of modern buoy data and historical in-situ data. Consider to shorten section 4.6 and move the main message to the discussion/conclusions section.

Figures:

- Figure 5: Consider to replace the contour plot with a colour-coded plot in Figure 11

- Figure 10: Consider adding histograms for the three month. It is very difficult to make out any changes other than seemingly random snow redistribution.

- Figure 11: Panels b and d are quite redundant. Consider showing W99 or difference to W99 instead

- Figure 11: The scale of the colorbars in panels c and d are slightly and unnecessary different

- Figure 14: Should the standard deviation (std) not be shown in both directions from the different snow depth values

- Figure 14: Consider spelling out level ice snow depth (SD) as the acronym is ambiguous with standard deviation (std)

---

## Referee Comment (RC2) · Anonymous Referee #2 · 31 Jan 2018

This paper incorporates a wealth of data measured from the Russian Sever expeditions to improve our historical knowledge of snow depth on sea ice, in particular in the marginal seas. The data used represent a massive effort spanning several decades and I am happy to see such a study done. The paper is thorough and generally well written, though I do have a few points I would like to see addressed.

An updated climatology to that produced by Warren et al., 1999 is one of the main results of the paper. However, in the Warren paper the Sever data were examined but not used for these reasons quoted in the paper:

"It is puzzling that the snow should be so much deeper around hummocks (45 cm) than behind ridges. The geographical patterns are also puzzling. Because some of the variation in average snow depth across the Arctic seen in Fig. 9 is probably due to different

areal coverages of sastrugi, ridges, and hummocks, one would expect the geographical gradients of snow within these classifications to be smaller than those of Fig. 9. However, this is not the case. The snow depth behind ridges appears to decrease toward Canada, while the height of snow around hummocks increases. These strange patterns cause us to question the representativeness of the measurements made at the aircraft landing sites. We favor the measurements made at the NP stations that were conducted more systematically."

I believe these points need to be addressed directly by the authors prior to publication.

In addition to the points raised in the Warren paper, I would like to see a better explanation for why the new climatology was produced using only the sastrugi and landing snow depth. The Warren climatology used data from snow lines which contains a mixture of snow depth from level ice as well as deformed ice, I'm not sure the snow depths produced from the Sever data would be equivalent. Perhaps a statistical analysis could be done to better relate data from the snow lines to that sampled in the Sever data.

Specific comments

P1 L23-24: In comparing to the Warren climatology it is necessary to state what is being compared: level ice snow or does it mix in snow from deformed ice too?

P3 L1-2: The Warren climatology gives a representation of the mean error in the form of the interannual variability, so I don't think this statement is correct here. The climatology did not have adequate sampling to provide information on the errors due to spatial variability, so I suggest this statement be revised to reflect this aspect.

P6 L20-34: In the W99 paper there is significant discussion about the representativeness of the sampling. While this section describes the sampling method, this important point has not been addressed. I note particular the analysis done with random samplings of the same population to see how the error for a given set of measurements changes with sample size and snow depth.

P7 L15-16: This statement seems like it belongs more in the caption for Figure 4.

Figure 5: It would be easier to read if the figure panel with snow depth had the units labeled.

P9 L19: I don't understand the last sentence, particularly with regard to the word "implying". Were all snow measurements used or was the MY subset used?

P10 and throughout: "fastice" should be "fast ice"

P13 L4-8: I'm confused by this section as the regression equation implies the sastrugi height is simply a constant 15.5 cm higher than the undisturbed snow.

P20 L5-8: Why were only the snow on the landing area and sastrugi data used and not any of the others described?

P20 L15-18: Although the detail of the data is lost in the quadratic fit, an advantage of the W99 climatology is that the fit coefficients were provided such that others could easily reproduce the climatology values. I suggest the authors put the fit coefficients in here.

P25 L34-35: Petty et al., 2016 (The Cryosphere) found FY feature heights of around 1 m which might be a more thorough comparison to the Sever data.

A number of minor grammar errors are present throughout the text.

———————————————

---

## Referee Comment (RC3) · Anonymous Referee #3 · 1 Feb 2018

The manuscript presents an analysis on the snow data collected during the Sever expeditions with the objective of producing an improved climatology over the historical climatology by Warren et al., 1999. The analysis provides useful statistics on the sampling scheme from the Sever expedition, and demonstrates relationships between snow depth and morphological features. However, there are critical limitations to the Sever snow data set, as addressed in Warren et al., 1999 (pages 1825-1827). These limitations were considered too biased to incorporate into the historical climatology. The manuscript's main conclusions overlook these limitations and over-interpret how representative the Sever data set is. More descriptions are needed on the assumptions made in the methodology and approach in the statistical analyses in consideration of these limitations. Please find specific comments below that I hope the authors will find

[Figure]

useful:

- In general, there's a lack of references throughout the manuscript; more references would help bolster the explanations of snow processes and interpretation of the results.

Page 2, Lines 15-30+. There have been numerous campaigns that have sampled snow on sea ice outside of the list presented here.

Page 3, Line 1. Figure 1 in Warren et al., 1999 shows that more than two stations were regularly present in a given year.

Page 3, Line 2. The instrumental errors were likely quite small. What do the authors mean here exactly?

Page 6, Lines 6-7. Snow lines were selected on a flat ice surface is contradictory to the description in Radionov et al., 1997 and Warren et al., 1999.

Page 6, Line 20. Please provide information on the spatial domain in which the 10-20 random snow thickness measurements were made.

Page 6, Lines 23-24. This indicates that the sampling was biased towards level first year sea ice, which is one reason to question how representative the Sever data set is.

Page 6, Lines 27-28. Does the 10 cm threshold for sample size introduce an additional bias to the data set?

Figure 4. What is meant by the prevailing landing area ice? Are these measurements from the runway or surrounding sea ice? Do these data also include sastrugi and ridge measurements?

Page 9, Lines 8-9. How was the 2.0 m threshold chosen, and how sensitive are the results to this threshold?

Page 9, Lines 13-14. It's not clear why only multiyear sea ice observations were included in the analysis here. Please provide more explanation on this decision, and

whether it is a valid assumption for creating an historical climatology for the Sever region.

Page 11, Line 3. Which measurements (ridge, sastrugi) are included in the average snow depth?

Page 11, Lines 16-17. This statement needs support (quantitative results) from a statistical analysis.

Page 12, Lines 6-8. It would be helpful to state that this relationship is dependent on the season, spatial domain, and sea ice type.

Figure 8. This figure doesn't show new information from Figure 16 in Warren et al., 1999.

Page 14, Lines 6-7. How much did spatial variation between landing sites affect the difference between the March and May values, rather than the conclusion that it's an increase? Were there equal sample sizes between the months of March, April, and May at the same sites?

Page 15, Lines 12-13. Is this a representative statistic if the landings were biased towards level first year sea ice?

Page 15, Lines 20-22. This is not correct. Multiyear ice has more variable surface relief, which acts to create more variability in snow depth distributions than level first year sea ice. Wind speed is not a valid explanation for the difference considering the observed frequency of blowing snow events.

Page 16, Lines 23-24. This finding is unclear.

Page 17, Line 1. The wind speed needs to be at least 5 m/s in order for snow to drift and redistribute. Wind speed is not a valid explanation for the difference.

Page 17, Lines 12-15. Where did these results come from if fast ice observations were not made during the Sever expeditions?

Page 20, Lines 5-9. Why were these adjustments made?

Figure 12. The buoy data need to be quality checked. Buoy 2013F does not show a realistic snowfall event.

---

## Author Comment (AC2) · 24 Mar 2018

The authors would like to thank the reviewer for his time and valuable comments. The corresponding changes and refinements have been made in the revised paper and are also summarized in our reply below. Authors' responses are in blue. Reviewer's comments are in black. When our manuscript is cited, it is shown in italics.

This paper incorporates a wealth of data measured from the Russian Sever expeditions to improve our historical knowledge of snow depth on sea ice, in particular in the marginal seas. The data used represent a massive effort spanning several decades and I am happy to see such a study done. The paper is thorough and generally well written, though I do have a few points I would like to see addressed.
An updated climatology to that produced by Warren et al., 1999 is one of the main results of the paper. However, in the Warren paper the Sever data were examined but not used for these reasons quoted in the paper:
"It is puzzling that the snow should be so much deeper around hummocks (45 cm) than behind ridges. The geographical patterns are also puzzling. Because some of the variation in average snow depth across the Arctic seen in Fig. 9 is probably due to different areal coverages of sastrugi, ridges, and hummocks, one would expect the geographical gradients of snow within these classifications to be smaller than those of Fig. 9. However, this is not the case. The snow depth behind ridges appears to decrease toward Canada, while the height of snow around hummocks increases. These strange patterns cause us to question the representativeness of the measurements made at the aircraft landing sites. We favor the measurements made at the NP stations that were conducted more systematically."
I believe these points need to be addressed directly by the authors prior to publication. In addition to the points raised in the Warren paper, I would like to see a better explanation for why the new climatology was produced using only the sastrugi and landing snow depth. The Warren climatology used data from snow lines which contains a mixture of snow depth from level ice as well as deformed ice, I'm not sure the snow depths produced from the Sever data would be equivalent. Perhaps a statistical analysis could be done to better relate data from the snow lines to that sampled in the Sever data.

In the Warren et al., 1999 (W99) paper the authors mention only data from Sever expeditions collected in April. However there were also quite sufficient number of measurements collected in March and May. In our paper we have processed observations collected in March, April and May (the MAM months) that indicates enlarging the amount of processed data in our case. It is difficult to say how larger the amount of data in our case is since in W99 there is no information about the number of processed observations. The Sever data set from NSIDC (that we use) contains snow depth measurements distributed over the MAM months in the following proportion: 46% of all measurements were done in April, 27% in March and 27% in May. Thus, March and May observations make a substantial addition to data collected in April. It is stated in W99 that geographical sampling of landings was very uneven, however the distribution of sampling sites used in the analysis is not shown and it is not possible to compare it with what was available for the analysis in our case. Contrarily to W99, we openly show the distribution of sampling sites and the number of processed observations. The suspect that we and W99 have processed different data sets is confirmed by comparing W99's estimates of the average snow depths behind pressure ridges and near hummocks. W99 indicates the "puzzling" (page 1826) fact that the estimates of the mentioned parameters are very different. In our case the parameters are comparable (see Table 4 and Fig. 9 of the paper) and their estimations are different from W99's. Decreasing snow

depth behind ridges towards Canada mentioned by W99 (page 1826) is obviously caused by the deficiency of data in Sever dataset collected in the region north of the Canadian Archipelago.
NP measurements were conducted on the MY ice and on the same floe throughout the life of the NP station. Snow line measurements catch very well natural variability of snow conditions on the spot, however, taking into account limited number of NP expeditions and unevenness of the distribution of their measurements in the Arctic, one may choose to find additional sources of data. Furthermore, one could also find strange geographical patterns in the W99 climatology: for example, according to W99 the snow depth in the area to the east of Greenland changed from 40 cm in March and April to 34 cm in May and in June it increased up to about 46 cm.
As to comparability of NP data and Sever data used in new climatology, there is no information about ice conditions, corresponded to NP snow line measurements, that complicates building a valid imitation. Ice conditions were not observed and did not described, so we do not know how much deformed ice affected the average snow depth measured along the lines. We read in W99: " The deep snow at about one-third of the way along the line in March **is probably** in a snow dune or in a drift near a pressure ridge" (Page 1817), which shows that there were no certainty about causes of snow depth variations among the authors.  We built the new climatology basing on the snow depth measured in the vicinity of landing site + the height of sastrugi, weighted in proportion to the sastrugi area. The height of snow attached to ice ridges was not included into calculations because we do not have estimations of the area covered by such features. The effect of not including that data in the computation results in some underestimation of the average snow depth.
In the paper:
*The height of snow attached to ice ridges was not included in the calculations because 1) the Sever measurements in the Western part of the Arctic Ocean are too scarce, and 2) there are no estimations of the area covered by such features from the Sever expeditions. The effect of not including that data results in some underestimation of the average snow depth.  The SHEBA observations indicated that in April and May 1998 about 3.9% of the examined area was covered by deep snow (>80cm) associated with ice ridges (Sturm et al., 2002). The underestimation is most important in the western Arctic, especially north of the Canadian Archipelago, where the highest concentration of the ridged ice is expected (Bourke and McLaren, 1992, Makshtas et al., 2003, Shoutilin et al., 2005).*

**Specific comments**
P1 L23-24: In comparing to the Warren climatology it is necessary to state what is being compared: level ice snow or does it mix in snow from deformed ice too?
The referee refers to the text in the Abstract. We updated that text:
*The main result of the study is a new snow depth climatology for the late winter using data from both the Sever expeditions and the North Pole drifting stations. The Sever snow depth measurements related to undisturbed snow and sastrugi are used in the computation. The height of snow accumulated near the ice ridges was not included in the calculations because those Sever measurements are unevenly distributed in the Arctic and are too scarce in some regions, besides, there are no estimations of the area covered by such features from the Sever expeditions. The effect of not including that data results in some underestimation of the average snow depth.*
All details are discussed in the paper.

P3 L1-2: The Warren climatology gives a representation of the mean error in the form of the interannual variability, so I don't think this statement is correct here. The climatology did not have adequate sampling to provide information on the errors due to spatial variability, so I suggest this statement be revised to reflect this aspect.
The sentence "*Due to the high spatial and temporal variability of the snow depth it is difficult to estimate the errors of the mean values of the W99 climatology.* "  is removed.

P6 L20-34: In the W99 paper there is significant discussion about the representativeness of the sampling. While this section describes the sampling method, this important point has not been addressed. I note particular the analysis done with random samplings

of the same population to see how the error for a given set of measurements
changes with sample size and snow depth.
We have done the analysis of sampling statistics. For every sea and for the central Arctic random
samplings of all available observations have been generated. The subsets contained 50, 100, 150, 200,
and more (where possible) observations with the increment 50. 100 subsets were generated for every
number of subset population in every region. Average snow depth was calculated for every subset
population. Variability between averages was estimated by standard deviation.

[Figure]

**Additional Figure**. Standard deviation of the average of a set of snow depth measurements as a function
of the number of landings used in the average computation. Measurements were randomly selected
from the whole set of data for every region. The amount of randomly generated subsets was 100 for
every number of sampled landings.

P7 L15-16: This statement seems like it belongs more in the caption for Figure 4.
This text describes what is shown in Fig.4. It is not the caption, just the text of the paper. The format
that we have to follow does not differentiate between the text of the figure caption and the text of the
paper.

Figure 5: It would be easier to read if the figure panel with snow depth had the units
labeled.
Please see the updated figure. We have tried to take into account all comments regarding it.

[Figure]

P9 L19: I don't understand the last sentence, particularly with regard to the word "implying".
Were all snow measurements used or was the MY subset used?

All measurements collected in the marginal seas were used:
"*By using the same ice thickness threshold in the marginal seas, the fraction of data coming from MY ice was 9% in the Kara Sea, 11% in the Laptev Sea, 34% in the East-Siberian Sea and 23% in the Chukchi Sea. These fractions of MY ice seem reliable, thus all snow observations in the marginal seas were used in the subsequent analysis*"  (rewritten)

P10 and throughout: "fastice" should be "fast ice"
Done.

P13 L4-8: I'm confused by this section as the regression equation implies the sastrugi height is simply a constant 15.5 cm higher than the undisturbed snow.
This result is a surprise for us too. Please keep in mind that it's an average over the whole dataset.

P20 L5-8: Why were only the snow on the landing area and sastrugi data used and not any of the others described?
Snow attached to ice ridges and hummocks is not included into calculation because there is no data describing their density. The updated text is shown here:
*In order to produce an integrated map that describe the average state of snow cover in the MAM months, the Sever and NP measurements have been combined by gridding all Sever data on snow depth on the level ice together with NP data. The height of sastrugi weighted in proportion to the sastrugi area was added for each Sever grid cell snow depth using the formula:  $H_{sev} = H_s + P_{sas}^{reg} \cdot H_{sas}$, where $H_s$ is the depth of snow measured on the level ice (described by data providers as the snow depth on the prevailing ice in the landing area), $H_{sas}$ is the height of sastrugi, and $P_{sas}^{reg}$  is the average portion of the ice surface covered by sastrugi in the region, to which snow measurement belongs (see Table 3). In the central Arctic, where the ice was mainly perennial at the time of measurements, only observations made on the MY ice during Sever expeditions have been included as was explained in Sect. 3. The height of snow attached to ice ridges was not included in the calculations because 1) the Sever measurements in the Western part of the Arctic Ocean are too scarce, and 2) there are no estimations of the area covered by such features from the Sever expeditions. The effect of not including that data results in some underestimation of the average snow depth.  The SHEBA observations indicated that in April and May 1998 about 3.9% of the examined area was covered by deep snow (>80cm) associated with ice ridges (Sturm et al., 2002). The underestimation is most important in the western Arctic, especially north of the Canadian Archipelago, where the highest concentration of the ridged ice is expected (Bourke and McLaren, 1992, Makshtas et al., 2003, Shoutilin et al., 2005).*

P20 L15-18: Although the detail of the data is lost in the quadratic fit, an advantage of the W99 climatology is that the fit coefficients were provided such that others could easily reproduce the climatology values. I suggest the authors put the fit coefficients in here.
The coefficients are calculated:

| Coeff | value |
|---|---|
| $H_0$ | 35.05 |
| 1 | -4.96E-06 |
| 2 | -1.46E-06 |
| 3 | -2.27E-12 |
| 4 | 2.91E-12 |
| 5 | -1.16E-12 |

1-5: coefficients in the regression equation. Snow depth: $H_s = H_0 + C_1*x + C_2*y + C_3*x^2 + C_4*xy + C_5*y^2$.
Coordinates are in meters. Projection is North Pole Stereographic, datum WGS_1984, latitude of origin = 90.0, central meridian = 20.0.
The text in the updated paper:

*The two-dimensional quadratic fit has been calculated as*

$H_s = H_0 + C_1 \cdot x + C_2 \cdot y + C_3 \cdot x^2 + C_4 \cdot xy + C_5 \cdot y^2$,

where $H_0 = 35.05$ cm, $C_1 = -4.69 \cdot 10^{-6}$, $C_2 = -1.46 \cdot 10^{-6}$, $C_3 = -2.27 \cdot 10^{-12}$, $C_4 = 2.91 \cdot 10^{-12}$, $C_5 = -1.16 \cdot 10^{-12}$, and x and y are coordinates in the North-Pole Stereographic projection. Units of x and y are meters.

P25 L34-35: Petty et al., 2016 (The Cryosphere) found FY feature heights of around 1 m which might be a more thorough comparison to the Sever data.
In Petty et al., 2016, the authors derive information regarding the heights of the sea ice topographic features using the elevation threshold of 20 cm. Though the snow elevations including sastrugi have to be captured by methodology, it is difficult to identify them and separate from the ice features.

A number of minor grammar errors are present throughout the text.

**Interactive comment on The Cryosphere Discuss., https://doi.org/10.5194/tc-2017-278, 2017.**

---

## Author Comment (AC3) · 24 Mar 2018

The authors would like to thank the reviewer for his/her time and valuable comments. The corresponding changes and refinements have been made in the revised paper and are also summarized in our reply below. Authors' responses are in blue. Reviewer's comments are in black. When our manuscript is cited, it is shown in italics. When other papers are cites, they are in green.

The manuscript presents an analysis on the snow data collected during the Sever expeditions with the objective of producing an improved climatology over the historical climatology by Warren et al., 1999. The analysis provides useful statistics on the sampling scheme from the Sever expedition, and demonstrates relationships between snow depth and morphological features. However, there are critical limitations to the Sever snow data set, as addressed in Warren et al., 1999 (pages 1825-1827). These limitations were considered too biased to incorporate into the historical climatology. The manuscript's main conclusions overlook these limitations and over-interpret how representative the Sever data set is.

In the Warren et al., 1999 (W99) paper the authors mention only data from Sever expeditions collected in April. However there were also quite sufficient number of measurements collected in March and May. In our paper we have processed observations collected in March, April and May (the MAM months) that already indicates enlarging the amount of processed data in our case. It is difficult to say how larger the amount of data in our case is since in W99 there is no information about the number of processed observations. The Sever data set from NSIDC (that we use) contains snow depth measurements distributed over the MAM months in the following proportion: 46% of all measurements were done in April, 27% in March and 27% in May. Thus, March and May observations make a substantial addition to data collected in April. It is stated in W99 that geographical sampling was very uneven, however the distribution of sampling sites used in the analysis is not shown and it is not possible to compare it with what was available for the analysis in our case. Contrarily to W99, we openly show the distribution of sampling sites and the number of processed observations. The suspect that we and W99 have processed different data sets is confirmed by comparing W99's estimates of the average snow depths behind pressure ridges and near hummocks. W99 indicates the "puzzling" (page 1826) fact that the estimates of the mentioned parameters are very different. In our case the parameters are comparable (see Table 4 and Fig. 9 of the paper) and their estimations are different from W99's. Decreasing snow depth behind ridges towards Canada mentioned by W99 (page 1826) is obviously caused by the deficiency of data in the region close to Canada.

Sever measurements were dependent on the availability of a landing spot. Landings were conducted in the areas where a proper runway was available. It might make an impression that the measurements were heavily biased to the level FY ice, however it was not so. As mentioned in the paper, the difference between the thickness of the runway ice and the ice of the area where landing measurements were conducted was in some cases about 300 cm and on the whole 46% of the measurements were conducted on the ice with the ice thickness larger than ice thickness of the runway. In April about 50% of all measurements were made on the ice with the thickness greater than 200 cm. Collected measurements of snow related to ice ridges and hummocks confirm that deformed ice was visited and surveyed, Fig. 3 shows the "intensity" of corresponded observations.

Comparing Sever data with NP data we would like to mention the following points. Snow observations in the MAM months available from the Sever expeditions were done at more than 2330 landings (P7, L7). The number of monthly averaged snow depths from NP stations for the same months was 143 and all data were collected on the MY ice. ("Only 499 snow lines were measured over the whole period of NP expeditions" - W99, page 1820). Sever expeditions collected data on both types of ice (FY and MY). Data collected in the marginal seas is certainly the main benefit that we can gain from Sever expeditions

observations. In the central Arctic the measurements were less dense comparing to other regions and since visited ice in the central Arctic was quite diverse (at least from the point of view of its thickness) those measurements provide inhomogeneous picture of the snow depth that is reflected in the final map (P.21, Fig.11). NP measurements being "conducted more systematically" (Page 1826, W99) are obviously biased to the thick ice and to the specific ice conditions that were to be met when the station was established. Additionally, snow depth observations on every NP station were conducted over the same line on the same floe that also limit representativeness of NP measurements. However, combining NP and Sever observations in the central Arctic, we believe that we merge data that complement each other and build the best possible picture of the Arctic ice in 1970-80s.

More descriptions are needed on the assumptions made in the methodology and approach in the statistical analyses in consideration of these limitations. Please find specific comments below that I hope the authors will find useful:

- In general, there's a lack of references throughout the manuscript; more references would help bolster the explanations of snow processes and interpretation of the results.
Which specific references does the referee have in mind? Please give us concrete references and we will happily mention them.

Page 2, Lines 15-30+. There have been numerous campaigns that have sampled snow on sea ice outside of the list presented here.
The updated text (with more references) :
*Valuable data on snow properties has also been collected from other expeditions, buoy measurements, ice camps and validation experiments in specific areas of the Arctic. In situ snow depth measurements has been carried out in spring time in the Beaufort Sea, Elson Lagoon and Chukchi Sea (Sturm et al., 2002, Sturm et al., 2006, Markus et al., 2006, Newman et al., 2012, Nghiem et al, 2013, Webster et al, 2014), in the Canadian waters between islands (Iacozza and Barber, 1999, King et al., 2015) and near the coast of northern Greenland (Farrel et al., 2012, King et al., 2015). With the aim to describe spatial distribution of snow on sea ice as a function of ice type, field experiments SIMMS'95 and C-ICE'96 in the Canadian Arctic have been conducted (Iacozza and Barber, 1999). Observations were made at the sites covered with only one sea ice type and snow topography on a given type of ice was described using variogram modelling. In 1997-98, extensive snow depth measurements have been made during SHEBA (Sturm et al., 2002). One of the objectives was to record temporal evolution of snow depth over the year, to evaluate its spatial variability (as broad as conditions of the experiment allowed), and to estimate the freshwater amount contained in the snow cover. Changes of snow depth connected with ice type and the level of deformation were also studied under SHEBA. Another expedition, the AMSR-Ice03 validation campaign carried out in March 2003 offshore of Barrow collected snow data for comparison with satellite products and also for analysis of snow depth on the sea ice of different age and different roughness (Sturm et al., 2006, Markus et al, 2006). During the Norwegian Young Sea ICE (N-ICE2015) campaign, snow depth was measured on the sea ice north of Svalbard (Gallet et al., 2017, Merkouriadi et al., 2017). In recent years a series of IceBridge validation/calibration campaigns have been conducted including in situ snow measurements on several ice types: undeformed level first-year (FY) ice, multiyear (MY) ice, and heavily deformed pressure ridges. Results have been published from studies near Greenland in April 2009 where measurements collected at the Danish GreenArc sea ice camp (Farrell et al., 2012), from surveys in the Beaufort Sea in March 2011 (Gardner et al., 2012; Newman et al., 2014), from measurements taken in March 2012 during BROMEX field campaign (Webster et al., 2014) and in March-April 2014 in the Canadian Arctic Archipelago waters close to Eureka and in the Lincoln Sea near the northern coast of Greenland (King et al., 2015). Assessment of five snow depth retrieval algorithms that differently process IceBridge snow radar data has been made through comparison with field*

*measurements from two ground-based campaigns, 2012 BROMEX near Barrow, Alaska, and 2014
Eureka, near the research base with the same name in Canada (Kwok et al., 2017).*

Please give us concrete references to add.

Page 3, Line 1. Figure 1 in Warren et al., 1999 shows that more than two stations were
regularly present in a given year.
In W99, Page 1820, it is twice mentioned that "typically (usually) only two stations were operating at
any time". The next figure, produced by co-authors confirms that. Points on the graph correspond to
months when snow line measurements were conducted in 1937-1987.

[Figure]

In the period from Nov.1968 to Dec.1971 there were 3 and even 4 stations conducting snow
measurements in the Arctic, but that was a very short period.

Page 3, Line 2. The instrumental errors were likely quite small. What do the authors
mean here exactly?
The authors did not mean instrumental errors. We rephrase the sentence as following:
" *Due to the high spatial and temporal variability of the snow depth it is difficult to adequately estimate
the uncertainty of the mean values of the W99 climatology*"

Page 6, Lines 6-7. Snow lines were selected on a flat ice surface is contradictory to
the description in Radionov et al., 1997 and Warren et al., 1999.
"*The snow line was selected on a flat ice surface with no human objects or ice hummocks that could
influence the snow depth (Colony et al, 1998)*."
Selection of snow lines is not described in Radionov et al., 1997 and Warren et al., 1999 explicitly. Only
Colony et al., 1998 describes the ice surface where the line was selected (see citations).
It is also worth to notice that in W99 it is written that " this dataset is described by Colony et al. (1998)".
**Colony et al., 1998:** " Snow-line measurements of the snow-cover characteristics have been carried out
since 1954. As a rule, the investigations were conducted once per month and sometimes once per 10-
day period. The sites of the snow-line measurements were chosen away from the ice station. **Sites were
selected on a flat ice surface and located clear of any constructions and ice hummocks**. The same snow
line was used for these measurements during the entire period of station operations."
**Warren et al., 1999:** " The direction of the **snow line was chosen randomly** when a station was
established, but once chosen, subsequent measurements were made along that same line for the
lifetime of the station. (Page 1816) ... *Later in the text*: When available, we use measurements on the
snow lines in preference to those at the stakes, for two reasons: 1) The snow lines cover a long enough
track to obtain a representative distribution of snow depths, passing through sastrugi, snow dunes, and
pressure ridges as well as level snow." (Page 1817) *and* "The deep snow at about one-third of the way
along the line in March **is probably** in a snow dune or in a drift near a pressure ridge." (Page 1817)
The remark "probably" shows that ice conditions corresponded to measured snow depths were not
known. In our paper we mention that point (P.19, L.13)
**Radionov et al., 1997**: "To determine the snow depth and density as well as the water equivalent of the
snow in the areas adjacent to the station, snow measurements were carried out along lines 1000 m in
length established outside the boundaries of the station. **The orientation of a line could be set as
desired relative to the structures at the station**, but once established, the lines were not changed
during the operation period of the station. " (Page 2-1).

We interpreted both Colony and Warren as following: the line was chosen on a flat ice surface, but later due to several natural processes ridges appeared.

Page 6, Line 20. Please provide information on the spatial domain in which the 10-20 random snow thickness measurements were made.
There is no information about that in the description of data.
Data description (from http://nsidc.org/data/g02140#title8) :
Before an ice landing, a characteristic site of ice- and snow-parameter variation was selected. Ice thickness and snow depth were evaluated at different locations on first-year, as well as multiyear ice…
Representative areas for measuring snow parameters were chosen from the air. After landing, snow depth was measured at 10-20 random points and on characteristic forms of ice surface terrain (including level ice, frozen melt ponds, and ridges). Snow depth on level first-year and, whenever possible, on multiyear ice was measured at 3-5 points on the runway. For snow cover more than 10 cm deep, at least 10 measurements were made over the entire ice floe, as well as on adjacent floes. Snow depth in 2-3 snow-covered ridges was measured on both windward and lee sides using a measuring pole at 10-20 points. The area covered by sastrugi was estimated from the air. After landing, the airborne observation data were checked, and other measurements made. Lengths and depths (at their mid-lengths) of snow dunes stretching from ice ridges at various angles were measured at 3-5 sites during every landing (Romanov 1995).

Page 6, Lines 23-24. This indicates that the sampling was biased towards level first year sea ice, which is one reason to question how representative the Sever data set is.
The next sentence (next to the sentence mentioned by referee) in the paper just says that the conditions around the landing place could be quite different from the conditions that characterize the runway: " *the difference between the ice thickness of the runway and of the area where other measurements were conducted was in some cases about 300 cm*"
We can add that, for example, in April about 50% of all measurements were done on the ice with the thickness greater than 200 cm; 46% of all measurements were conducted on the ice with the ice thickness different (larger) than the ice thickness of the runway.

Page 6, Lines 27-28. Does the 10 cm threshold for sample size introduce an additional bias to the data set?
In the Sever data description (see the data description above) they specifically mention situation when snow thickness was more than 10 cm, however it does not mean they did not measure smaller snow depths. In the data set there are a lot of measurements showing snow thickness smaller than 10 cm.

In order to make the matter more clear we have added some clarifications in the description of data:
*The runway was chosen on flat ice that was most probably first year ice, but could also be multiyear ice. Meanwhile, the ice conditions around landing track were usually different from that on the runway: the difference between the ice thickness of the runway and of the area where other 25 measurements were conducted was in some cases about 300 cm. In the description of data the ice in the area around runway is called "prevailing ice of the landing area ". Later in the paper we use definition "undeformed ice" as a substitute for "prevailing ice of landing area", since snow cover associated with ice features caused by ice deformation like ice ridges and hummocks is described separately. After landing, snow depth was measured at 10-20 random points on prevailing ice of the landing area and on ice surface with distinctive features. For snow depth of more than 10 cm, at least 10 measurements were conducted over the entire ice floe, as well as on adjacent floes. How many measurements were made in case when snow depth was lower than 10 cm is not indicated in the description of data. The depth of snow dunes stretching from ice ridges and depth of snow on hummocks were measured using the following steps. The snow depth on 2 or 3 snow-covered hummocks was measured on both windward and lee sides at 10-20 points. The depth of snow dunes stretching from ice ridges were measured at 3-5 sites at their mid-length. The height of sastrugi on the undeformed ice was measured at several points. Note that all types of snow dunes formed on a flat ice surface by wind were referred to as sastrugi in Sever expeditions' data set. The averaged measurements of the mentioned parameters were reported in the documents from each expedition.*

Figure 4. What is meant by the prevailing landing area ice? Are these measurements from the runway or surrounding sea ice? Do these data also include sastrugi and ridge measurements?

The combination of words "prevailing landing area ice" was used in the description of data (by data providers). That's why we use it too. It is surrounding undeformed ice. In the paper, we use definition "undeformed ice" as a substitute. In cases when there were ice features like ridges and hummocks around, snow measurements were made too and depth of snow associated with those features was reported separately. Sastrugi (formed on the flat ice surface due to wind action) were measured and reported separately. See updated text *in Italic* above. In the paper the Figure 4 is mentioned in the sentence (P.7, l.14-15):

*Data on the depth of undisturbed snow measured on the prevailing type of ice in the landing area from all Sever expeditions landings over the period from 1959 to 1988 is presented in Fig. 4.*

Page 9, Lines 8-9. How was the 2.0 m threshold chosen, and how sensitive are the results to this threshold?

In the "WMO Sea-Ice Nomenclature. WMO - No.259. Volume 1 – Terminology and Codes" FY ice is described as "Sea ice of not more than one winter's growth, developing from *young ice*; thickness 30 cm - 2 m. May be subdivided into *thin first-year ice/white ice*, *medium first-year ice* and *thick first-year ice*."

In order to make the matter more clear we rewrote the paragraph on P.9, L. 8-14:

*In the analysis it was useful to discriminate observations from FY and MY ice because MY ice as a platform to accumulate snow is different from FY ice. Firstly, MY ice exists from the very beginning of the fall and thus is able to catch the earliest falling snow. Secondly, the topography of MY ice is different from that of FY ice, being more irregular, that influences redistribution of snow. We separate observations from FY and MY ice by using an ice thickness threshold, since ice thickness was measured at every landing. The threshold was chosen to 2.0 m, implying that ice thinner than 2 m is defined as FY ice and ice thicker than 2 m is defined as MY ice (WMO Sea-Ice Nomenclature). In the decades of the Sever expeditions, MY ice dominated in the central Arctic, the fraction of MY ice being close to 100% in 60s-80s. Using the 2 m threshold, it was found that 78% of Sever observations from the central Arctic were conducted on the MY ice and 22% on the FY ice. The sampling made by the Sever expeditions was probably biased towards level ice. In order to derive maximally representative data set for the central Arctic, we decided to use only MY-based measurements for that area. Supposing that NP and Sever observations complement each other we merged Sever snow depth observations from MY ice with NP data to produce snow depth climatology for the central Arctic.*

We understand that the threshold 2 m does not work in all cases, however we think its use will classify most of FY and MY ice correctly.

Page 9, Lines 13-14. It's not clear why only multiyear sea ice observations were included in the analysis here. Please provide more explanation on this decision, and whether it is a valid assumption for creating an historical climatology for the Sever region.

The updated text is as follows:

*In the analysis it was useful to discriminate observations from FY and MY ice because MY ice as a platform to accumulate snow is different from FY ice. Firstly, MY ice exists from the very beginning of the fall and thus is able to catch the earliest falling snow. Secondly, the topography of MY ice is different from that of FY ice, being more irregular. We separate observations from FY and MY ice by using an ice thickness threshold, since ice thickness was measured at every landing. The threshold was chosen to 2.0 m, implying that ice thinner than 2 m is defined as FY ice and ice thicker than 2 m is defined as MY ice (WMO Sea-Ice Nomenclature). In the decades of the Sever expeditions, MY ice dominated in the central Arctic, the fraction of MY ice being close to 100% in 60s-80s. Using the 2 m threshold, it was found that 78% of Sever observations from the central Arctic were conducted on the MY ice and 22% on the FY ice. The sampling made by the Sever expeditions was probably biased towards level ice. In order to derive maximally representative data set for the central Arctic, we decided to use only MY-based measurements for that area. Supposing that NP and Sever observations complement each other we merged Sever snow depth observations from MY ice with NP data to produce snow depth climatology for the central Arctic.*

Page 11, Line 3. Which measurements (ridge, sastrugi) are included in the average snow depth?
The average snow depth discussed here is the average snow depth shown in Table 2 (P.10, just before the text). In the Table header the parameters is described: "*Snow depth (SD) of undisturbed snow cover on level ice in the landing areas in different parts of the Arctic Ocean for the months March, April and May.*"
Snow attached to ridges and sastrugi is not included in these measurements. Those observations are described later in the paper.

Page 11, Lines 16-17. This statement needs support (quantitative results) from a statistical analysis.
We removed the mentioned statement from the place where it was.

Page 12, Lines 6-8. It would be helpful to state that this relationship is dependent on the season, spatial domain, and sea ice type.
The part of the paper related to the mentioned relationship has been rewritten:
*The start time of snow accumulation is one of the major factors determining the snow depth by the end of snow accumulation season (Radionov et al., 1997; Hezel et al., 2012). In the case of FY ice, snow accumulation can only start after the sea ice freezing is stable. A delayed sea ice freeze-up will lead to a delayed start of snow accumulation and thereby have impact on the snow depth evolution during the winter (Webster et al., 2014). MY ice begins to accumulate snow earlier in the fall; additionally, in some regions the snow can survive through the summer season. The ice thickness generally increases throughout the cold season, and though the sea ice growth rate is known to be inversely proportional to its thickness (e.g., L'Heveder and  Houssais 2001, Bitz and Roe, 2004) we would expect  the ice thickness to be related to the depth of snow accumulated during the winter. Figure 7 plots the depth of snow versus the ice thickness. The data are averages corresponded to ice thicknesses divided into 20 cm-ice-thickness intervals. All measurements used here were conducted throughout the Arctic (see Fig. 5a) in the MAM months on the undeformed ice.  The number of measurements is shown by the histogram. An empirical linear relation between ice thickness and snow depth can be derived for the FY ice, using least square regression, as shown in Fig. 7 by red line. The derived relation is the following:*
$H_s = 0.069 * H_i + 2.0.$
*In the equation $H_s$ is the snow depth of the undisturbed snow on the undeformed ice and $H_i$ is the ice thickness (both in cm). The FY ice has been separated using a threshold of 200 cm. The linear regression was carried out using averaged snow depth and mean ice thickness for each 20 cm-ice-thickness group of data. The coefficient of determination ($R^2$) is 0.95. The relation between snow depth and sea ice thickness for the whole data set can be described by the polynomial function (see Fig. 7). The highest values of ice thickness from the lowest number of measurements were rejected from the polynomial calculation.*

[Figure]

**Figure 7**. *The relation between the thickness of the undeformed ice and the depth of accumulated snow in the end of winter.*

Figure 8. This figure doesn't show new information from Figure 16 in Warren et al., 1999.
As we explained earlier (in the very beginning of this text) we process data from March, April and May, which is already new information with respect to Figure 16 in W99. Besides, we show distribution and amount of measurements (together with measured values). We think it all is different from isopleths based on April measurements shown in W99. The distribution of measurements (see Fig.8) shows that there were very little measurements related to hummocks and ridges in the Canadian Arctic and in the Beaufort Sea.
The questioning in W99 "The snow depth behind ridges appears to decrease toward Canada, while the height of snow around hummocks increases. These strange patterns cause us to question the representativeness of the measurements made at the aircraft landing sites." is based on comparison of regions with very different data sufficiency.

Page 14, Lines 6-7. How much did spatial variation between landing sites affect the difference between the March and May values, rather than the conclusion that it's an increase? Were there equal sample sizes between the months of March, April, and May at the same sites?
Measurements were never done at the same sites, at least there was no such task. The decision about landing had been taken basing on what the researches saw during the flight. " *By comparing monthly mean values for the MAM months, the height of sastrugi on the MY ice had increased from 31 cm in March to about 38 cm in May.* " - That's a statistical conclusion based on all data collected on the MY ice.

Page 15, Lines 12-13. Is this a representative statistic if the landings were biased towards level first year sea ice?
The landings were biased towards level FY sea ice, but the MEASUREMENTS were NOT.

Page 15, Lines 20-22. This is not correct. Multiyear ice has more variable surface relief, which acts to create more variability in snow depth distributions than level first

year sea ice. Wind speed is not a valid explanation for the difference considering the observed frequency of blowing snow events.

" *Winds in the central Arctic are weaker than in the Siberian seas and wind speed is less variable (Frolov et al., 2005, Martin et al., 2014), which gave a smoother snow depth histogram for the central Arctic in comparison to marginal seas (Fig. 9)*." We believe that wind has its influence. We added in the text of paper:

*Additionally, greater roughness of the MY ice surface exhibits more barriers for the blowing snow and aligns diversities.*

Page 16, Lines 23-24. This finding is unclear.
The rewritten sentence is the following:
*In the Kara Sea, there was the highest number of measurements comparing to other seas that probably allowed to observe most cases of snow dunes layouts and states.*

Page 17, Line 1. The wind speed needs to be at least 5 m/s in order for snow to drift and redistribute. Wind speed is not a valid explanation for the difference.
Analysis of NCEP Reanalysis Wind Product shows that in the MAM month in the Chukchi Sea the wind speed is higher than 5 m/s and higher than in other Siberian seas and in the central Arctic. See example.

[Figure]

It is also supported by the illustrations in the mentioned paper (Martin et al., 2014). Note, that in the text of paper we consider "*insufficiency of measurements*" as a possible cause as well.

Page 17, Lines 12-15. Where did these results come from if fast ice observations were not made during the Sever expeditions?
In order to make the matter more clear we changed the sentence slightly:
*In these cases the ice can become highly deformed with shear ridges or stamukhi (Barry et al., 1979). The areas of stamukhi were not observed by Sever expeditions because landing was not possible.*

Page 20, Lines 5-9. Why were these adjustments made?
The text (Page 20, from Line 4 onwards) has been changed:

*NP drifting station snow observations were not accompanied by any ice observations. Therefore the variability in NP snow depth measurements cannot be explained from different ice characteristics. That complicates processing Sever data with the aim to make them similar to NP data. The only information about NP snow measurements is that the NP data were collected from a solid MY floe. Figure 10 represents snow line observations made by NP16 drifting station personnel in 1969 in the MAM months. It shows that spatial variability of measured values was very high. The range of values over the same line was from 2 to 64 cm in March, from 4 to 90 cm in April and from 5 to 90 cm in May. The low correlation between repeated measurement lines (0.18 between March and April and 0.04 between April and March) suggests that the variations were primarily caused by wind.*

*In order to produce an integrated map of snow depth on the sea ice that describe the average state of snow cover in the MAM months, the Sever data and the NP observations have been combined in the following way. From all Sever observations data on snow depth on the level ice were used to make the base for gridding. That data were adjusted by adding the height of sastrugi weighted in proportion to the sastrugi area. The Sever snow depth for gridding in a given region is calculated using the formula: $H_{sev} = H_s + P_{sas}^{reg} \cdot H_{sas}$, where $H_s$ is the depth of snow measured on the level ice (described by data providers as the snow depth on the prevailing ice in the landing area), $H_{sas}$ is the height of sastrugi, and $P_{sas}^{reg}$ is the average portion of the ice surface covered by sastrugi in the region, to which snow measurement belongs (see Table 3). Sastrugi height has been added to snow depth on the level ice using the relation between $H_{sas}$ and $H_s$. In the central Arctic, where the ice was mainly perennial in the time of measurements, only observations made on the MY ice during Sever expeditions have been included as was explained in Sect. 3. The height of snow attached to ice ridges was not included into calculations because we do not have estimations of the area covered by such features. The effect of not including that data in the computation results in a slight underestimation of the average snow depth.*

*The snow depth distribution on the sea ice in the MAM months in 1960-80s has been generated using 2331 points, 143 of which were monthly averaged NP data and others were Sever snow depths calculated as described in the previous paragraph. The distribution of these points is presented in Fig. 11a.*

Figure 12. The buoy data need to be quality checked. Buoy 2013F does not show a realistic snowfall event.

That buoy worked another year and it produced reasonable results after the period shown in the paper (that raised doubts from the referee). Ref. : http://imb-crrel-dartmouth.org/imb.crrel/irid_data/2013F_thick.png   We believe that buoy's steady work in the later (following) year confirms its  normal functionality in the MAM months 2013.

**Interactive comment on The Cryosphere Discuss., https://doi.org/10.5194/tc-2017-278, 2017.**

---

## Author Response (AR1)

**Responses to the reviews, including relevant changes made in the manuscript, and a marked-up manuscript version**

The authors would like to thank all referees for the review of the manuscript, valuable comments and the discussion involved in the process. The corresponding changes and refinements have been made in the revised paper and are also summarized in our reply below. Authors' responses are in blue. Reviewer's comments are in black. When our manuscript is cited, it is shown in italics (black). When other papers are cites, they are in green.

**Response to S. Hendricks (Referee)**

The study "Snow depth on Arctic sea ice from historical in situ data" evaluates observations of snow and sea ice surface properties from the Sever aircraft landings on Artic sea ice from the 60th through 80th of the 20st century. The data contains snow depth from surfaces representative of the area near the landing site as well as some information on the snow depth distribution, from which the authors deduce average snow depth. This information is used to construct a climatological snow depth map based on significantly improved observational density in the Russian part of the Arctic compared to the Warren climatology. However, this climatology is only valid for the month between March and May due to visibility constraints for the landings.

The necessity and value to document and utilize these extensive observational datasets from the past cannot be understated and the paper constitutes a valuable contribution to this effort. The paper is generally well written and adds a thorough analysis to the documentation of the methodology.

We thank the reviewer for his overall positive evaluation of our work and constructive comments, which were of great help in revising the manuscript. The point by point response to the comments is listed below.

There are however a few general minor points and specific comments where the analysis and the presentation of the results could be improved before publication:

1) The authors provide a detailed climatology of average snow conditions but without a magnitude of the snow depth variability. It would be important to have this information as a measure the uncertainty of the climatology. Of course, variability can only be estimated in areas with repeated observations, but might be possible with pooling data on the Russian shelves.

We calculated standard deviation of the data used for producing the new climatology. It is calculated as a weighted standard deviation from variances of the snow depth and the height of sastrugi with a weight of 0.35, which is an average portion of sastrugi area in the Arctic. It is included in the updated Fig. 11 (see figure below, on the 3-rd page). Certainly, it is only a part of the uncertainty of the new climatology; however it is difficult to evaluate errors from other sources. First of all, the Sever data provided by NSIDC does not have any description of errors or uncertainty. Referring to our calculation, we have to note that the height of snow attached to ice ridges was not included in the calculation because the Sever measurements in the Western part of the Arctic Ocean are too scarce, and there are no estimations of the area covered by such features from the Sever expeditions. The effect of not including that data results in some underestimation of the average snow depth. The underestimation is most important in the western Arctic, especially north of the Canadian Archipelago, where the highest concentration of the ridged ice is expected.

2) The authors also did not show the difference to the Warren climatology. The improvements on the Russian shelves are obvious, but it would be valuable to assess the impact of the localized nature of the NP observations compared to the regional coverage of the Sever program on the generation of climatologies in regions where the observations should be comparable.

Below please see the mapped difference between the Warren climatology (W99) and the new one. It demonstrates the deficiencies of both approaches. On the one hand, it reveals artificiality of the smoothness of Warren's climatology - it is a consequence of estimating parameters through polynomial fitting in the areas where the values of those parameters change considerably. On the other hand, it shows that Sever expeditions did not provide enough data to get a smooth distribution in the central Arctic and to describe adequately snow depth in the area near the Canadian coast. Thus we have unphysical smoothness of W99's data and unevenness of the Sever data in some regions.

It is also worth to keep in mind that Warren's monthly climatology depends on the distribution of the available for that months NP measurements. Luck of measurements in some cases causes strange results: for example, according to W99 the snow depth in the area to the east of Greenland changed from 40 cm in March and April to 34 cm in May and in June it increased up to about 46 cm.

The difference between the Warren's snow depth and the new climatology based on Sever observations. The units are cm.

Below are W99 maps of snow depth for the MAM months.

---

## Author Response (AR2)

We thank the reviewer for his/her positive evaluation of the revised manuscript and additional comments. The point by point response to the comments is listed below. Authors' responses are in blue. Reviewer's comments are in black. When the updated manuscript is cited, it is shown in black italic.

**Page 2, Line 4**. Please provide a reference for the albedo value for snow. Contemporary studies have shown 0.85 for cold, dry snow.
Thank you for this correction. The updated text is the following:
*In winter the snow ensures high sea ice surface albedo of about 0.85 (Perovich and Polashenski, 2012) associated with low energy absorption.*
The reference is added to the reference list.

**Page 15, Lines 3-4**. It might be insightful to compare these values to those in Filhol and Sturm (2015).
The paper by Filhol and Sturm (2015) deals mainly with snow barchan/dune morphology revealing relationships between their width, length and height and presenting the model of snow bedform formation and dynamics. The measurements used in the study have been made in tundra and on the lakes (in winter). The formation of sastrugi is discussed and different forms are analyzed. However we do not see in that study any measurement statistics or modeled data that can be compared with our results.

**Page 16, Lines 23-24**. This explanation is not correct. The snow depth distributions in Figure 9 show that the Central Arctic has the largest variability relative to other regions (e.g., it has the widest spread and longest tail among the distributions). It only takes one wind event to redistribute the snow to the ice surface topography. Thus the magnitude of wind speed in different regions is not a convincing explanation for differences in snow drifts surrounding hummocks and ridges. If the authors choose to keep this statement in, I encourage them to change the wording to reflect that this is speculation:
"...which may have given a smoother snow depth distribution for the central Arctic."
The text has been changed:
*Average values of the hummock snow depth are highest in the central Arctic (Table 4) that can be expected. Differences in average snow depths on hummocks in the Siberian seas can be the result of selectiveness of measurements. The histogram of snow depth around hummocks for the central Arctic is the smoothest in comparison to marginal seas (Fig. 9). Asymmetry in distributions of the snow depth on hummocks in the seas may be the result of stronger and changeable winds and also perhaps a consequence of the movement of the floes.*

**Page 19, Figure 9**. There is a typo for "Chukchi Sea" in the bottom right panel.
Thank you for noticing that! The figure is corrected.

**Page 24, Line 13**. I suggest including a statement that notes the unrealistic jump in snow depth in IMB buoy 2013F. It is not possible for a ~25 cm snowfall event to occur within a four-hour time period.
The text has been updated. We added the following sentence:
*We should note that a ~25 cm increase of snow depth within a very short time period registered by the buoy 2013F seems unrealistic.*

**Page 27, Lines 28-29**. Were all IMB buoys included in this comparison for the Central Arctic? Some of the buoys were located in the marginal seas in Figure 12, which would be worth noting.

Only measurements from the buoys traveled in the Central Artic have been used for that estimation. The text has been updated:

*The average snow depth on the MY ice in the central Arctic for the MAM period is 24.3±0.7 cm according to recent IMB buoy measurements (from four buoys 2011M, 2012G, 2012J, 2015F, see Fig. 12) and 21.2±9.4 cm according to AWI snow buoy measurements.*

I do have **one final comment** that should be addressed: The equation for the quadratic fit is now given, however there is no time dependence. Is this the same for all months March, April, and May? Or are the coefficients different for the three months?

The quadratic fit has been calculated using measurements from all three months. We provide one set of coefficients for the MAM months. The text has been updated (Page 22, Line 14):

[revised manuscript text omitted]